# Aggregative *trans*-eQTL analysis detects trait-specific target gene sets in whole blood

Diptavo Dutta [1], Yuan He[2], Ashis Saha[3], Marios Arvanitis[2,4], Alexis Battle [2,3,6✉] & Nilanjan Chatterjee[1,5,6✉]

Large scale genetic association studies have identified many trait-associated variants and understanding the role of these variants in the downstream regulation of gene-expressions can uncover important mediating biological mechanisms. Here we propose ARCHIE, a summary statistic based sparse canonical correlation analysis method to identify sets of gene-expressions trans-regulated by sets of known trait-related genetic variants. Simulation studies show that compared to standard methods, ARCHIE is better suited to identify "core"-like genes through which effects of many other genes may be mediated and can capture disease-specific patterns of genetic associations. By applying ARCHIE to publicly available summary statistics from the eQTLGen consortium, we identify gene sets which have significant evidence of *trans*-association with groups of known genetic variants across 29 complex traits. Around half (50.7%) of the selected genes do not have any strong *trans*-associations and are not detected by standard methods. We provide further evidence for causal basis of the target genes through a series of follow-up analyses. These results show ARCHIE is a powerful tool for identifying sets of genes whose *trans*-regulation may be related to specific complex traits.

[1] Department of Biostatistics, Johns Hopkins University, Baltimore, MD, USA. [2] Department of Biomedical Engineering, Johns Hopkins University, Baltimore, MD, USA. [3] Department of Computer Science, Johns Hopkins University, Baltimore, MD, USA. [4] Department of Cardiology, Johns Hopkins University, Baltimore, MD, USA. [5] Department of Oncology, Johns Hopkins University, Baltimore, MD, USA. [6] These authors contributed equally: Alexis Battle, Nilanjan Chatterjee. ✉email: ajbattle@jhu.edu; nilanjan@jhu.edu

Genome-wide association studies (GWAS) have identified tens of thousands of common variants associated with a variety of complex traits[1] and a majority of these identified trait-related variants are in the non-coding regions of the genome[2–4]. It has been shown that these GWAS identified variants have a substantial overlap with expression quantitative trait loci (eQTL) i.e., variants that are associated with the expression levels of genes[5–7]. A number of tools[8–11] have been developed to identify potential target genes through which genetic associations may be mediated by investigating the effect of variants on local genes (cis-eQTL), typically within 1 Mb region around the variant, but underlying causal interpretation remains complicated due to linkage disequilibrium and pleiotropy. A recent study has shown that a modest fraction of trait-heritability can be explained cis-mediated bulk gene-expressions[12], but future studies with more cell-type specific information has the potential to explain further.

Compared to cis-eQTL, studies of trans-eQTL have received less attention though they have the potential to illuminate downstream genes and pathways that would shed light on disease mechanism. A major challenge has been the limited statistical power for detection of trans-eQTL effects due to much weaker effects of SNPs on expressions of distal genes compared to those in cis-regions and a very large burden of multiple testing. However, trans-effects, when detected, has been shown to be more likely to have tissue-specific effects[13,14] and are more enriched than cis-eQTLs among disease loci[15]. Trans-eQTLs are, in general, known to act on regulatory circuits governing broader groups of genes[16] and thus have the potential to uncover gene networks and pathways consequential to complex traits[17,18]. Limited studies of trans-eQTL effect of known GWAS loci have identified complex downstream effects on known consequential genes for diseases[15,19]. In fact, an "omnigenic" model of complex traits has been hypothesized under which a large majority of genetic associations is mediated by cascading trans-effects on a few "core genes"[20,21]. Thus, given the increasing scope of eQTL studies, it has become even more important to comprehensively identify trait-specific trans-associations to highlight biological processes and mechanisms underlying phenotypic change. However, to the best of our knowledge, no framework has been developed to detect such trait-specific trans-association patterns and sets of trait-relevant genetically regulated genes, specifically leveraging summary statistics from transcriptomic studies, which are more readily available than individual-level genotype data.

In this work, we propose a summary-statistics based method using sparse canonical correlation analysis (sCCA)[22–27] framework, termed Aggregative tRans assoCiation to detect pHenotype specIfic gEne-sets (ARCHIE), which identifies sets of distal genes whose expression levels are trans-associated to (or regulated by) groups of GWAS SNPs associated to a trait. The method requires summary statistics from standard SNP-gene expression trans-eQTL mapping, estimates of linkage disequilibrium (LD) between the variants and co-expression between genes, which can be estimated using publicly available datasets. Together, the selected variants and genes (jointly termed ARCHIE components) reflect significant trait-specific patterns of trans-association (Fig. 1A shows an illustration for the functionality of ARCHIE). Compared to standard trans-eQTL mapping, the proposed method improves power for detection of signals by aggregating multiple trans-association signals across GWAS loci and genes. Moreover, we propose a resampling-based method to assess the statistical significance of the top components of sCCA for testing enrichment of trait-specific signals in the background of broader genome-wide trans-associations. If multiple ARCHIE components are significant, they reflect approximately orthogonal patterns of trans-associations for the trait-related variants, with the selected target genes pertaining to distinct downstream mechanisms of trans-regulation.

Here we apply the proposed method to analyze large-scale trans-association summary statistics for SNPs associated with 29 traits reported by the eQTLGen consortium[19]. The results show that ARCHIE can identify trait-specific patterns of trans-associations and relevant sets of variants and co-regulated target genes. The majority (50.7%) of the target genes we detect, are novel, meaning they would not have been identified by standard trans-eQTL mapping alone. We provide independent evidence supporting our results, using a series of downstream analysis to show that the selected target genes are enriched in known trait-related pathways and define directions of associations for the SNPs that are more enriched for underlying trait heritability than expected by chance. The proposed methods can be further applied in the future to association statistics data on other types of high throughput molecular traits, such as proteins and metabolites, to understand their mediating role in genetic architecture of complex trait.

## Results

**Overview of methods**. We assume that we have summary statistics data (Z values and p values) available for a set of variants identified through large-scale GWAS of a given trait of interest, from standard trans-eQTL analysis across large number of distal genes. We further assume that we have additional reference datasets to estimate correlation (linkage disequilibrium) among the SNPs and among gene expressions in the underlying population of interest. ARCHIE uses these datasets to employ a sCCAs[22,24] which produces sparse linear combinations of the trait-related variants (termed variant-component) that is associated with sparse linear combination of genes (termed gene-component) where each non-zero element of the variant (or gene)-component indicates that the respective variant (or gene) is selected. (Fig. 1A shows a toy example of ARCHIE's functionality). The selected genes reflect sets that are broadly trans-regulated by the selected SNPs and via which the effects of selected SNPs on the trait appear to be mediated. Further, we evaluate whether the genes and variants selected in the ARCHIE components reflect significant trait-specific trans-association patterns, through a resampling method by comparing the observed sparse canonical correlation values to that expected from trans-associations of GWAS variants not specific to a trait (for details see Methods and Supplementary Note 1). We show through simulation and resampling studies that by jointly analyzing multiple GWAS variants associated to a trait, ARCHIE can identify broader downstream trans-regulatory mechanisms relevant to the trait compared to standard trans-eQTL mapping which identifies general trans-associations that might not be trait-specific and can arise due to factors like pleiotropy, correlated expressions, and others (See Supplementary Note 2).

**Simulation study results: comparison with standard trans-eQTL analysis and assessing trait-specificity**. We first compare ARCHIE with standard trans analysis using a series of simulation studies. ARCHIE addresses a composite null hypothesis that there is no association between a group of variants and genes, while standard trans-eQTL mapping tests a series of individual null hypotheses of no association across variant-gene pairs. Thus, these two methods are not directly comparable in terms of power unless the type-I error rates are calibrated with respect to a common benchmark. We first evaluate the type-I error rate for each method under a global null hypothesis of no association between a group of SNPs and a gene network (Fig. 2A). Type-I error rate for ARCHIE is defined as the proportion of simulation iterations where the p value for at least one ARCHIE component is less than the chosen α-level. Correspondingly, for standard trans-eQTL analysis, type-I error is calculated as the proportion

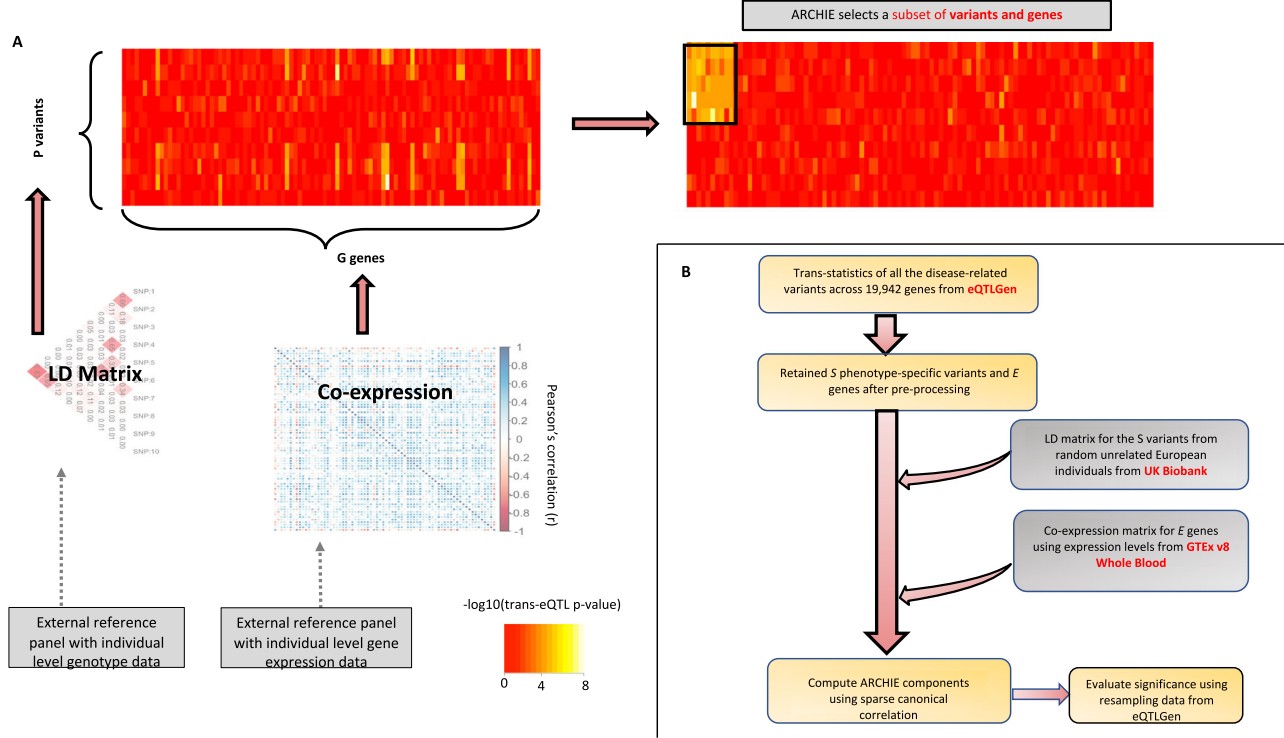

**Fig. 1 Overview of ARCHIE and Analysis pipeline. A** An illustrative example for ARCHIE. Association statistics ($-\log_{10} p$ value) from trans-eQTL mapping for *P* variants and *G* genes are shown in the left panel heatmap. Using LD and co-expression estimated from reference datasets, ARCHIE aggregates multiple weaker trans-eQTL associations to select a subset of variants and genes which capture trait specific trans-association patterns (right panel heatmap). **B** Description of analysis pipeline for summary statistics of trans-eQTL mapping provided by the eQTLGen consortium by integrating external estimates from GTEx (v8) and UK Biobank. See Results and Methods for more details.

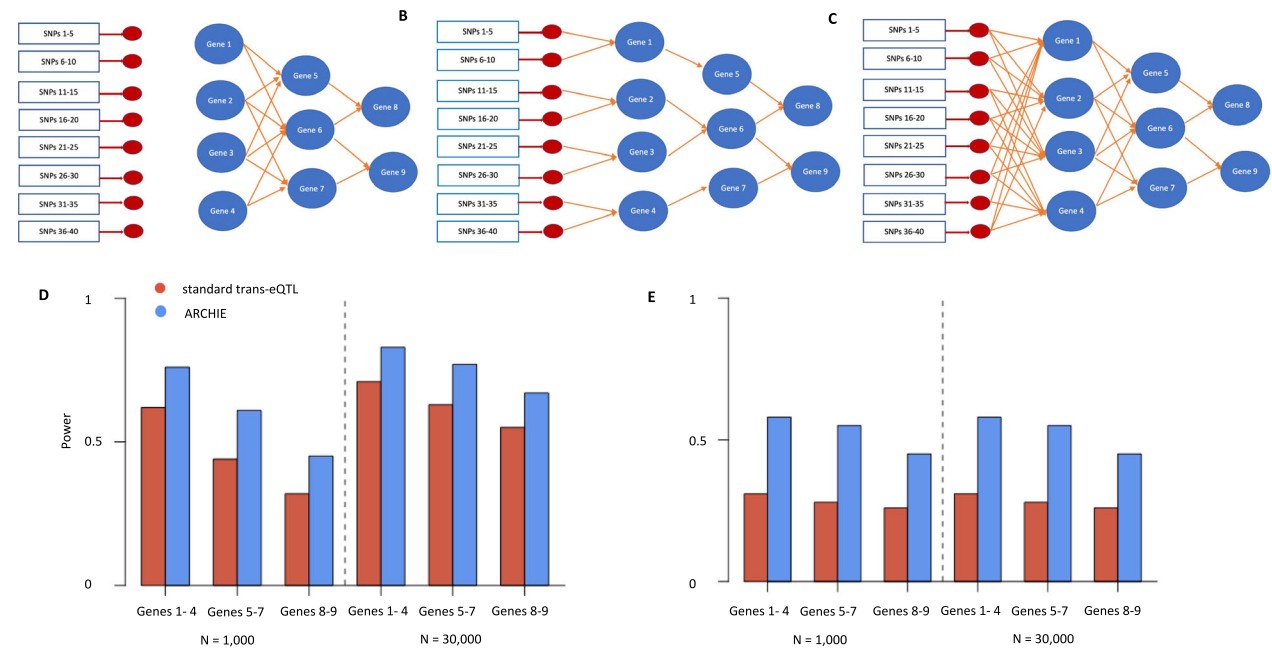

**Fig. 2 Simulation model and results. A** causal network model for global null scenario with no association between the SNPs 1-40 and the Genes 1-9 (marked in blue). See Results and Methods: Simulation Model for more details. **B** Sparse causal network: causal regulatory network of downstream Genes 1–9 (in blue) regulated by SNPs 1–40 mediated via 8 local (cis) genes (marked in red). **C** Dense causal network: causal regulatory network of downstream Genes 1–9 (in blue) regulated by SNPs 1–40 mediated via 8 local (cis) genes (marked in red). For both **B**, **C** the heritability of cis gene expressions explained was maintained at 20–22% and the heritability of downstream genes (1–9) explained was maintained at 10–14%. **D** Empirical power to identify different gene sets within sparse causal regulatory network (**B**) using ARCHIE compared to standard trans-eQTL mapping. **E** Empirical power to identify different gene sets within dense causal regulatory network (**C**) using ARCHIE compared to standard trans-eQTL mapping.

**Table 1 Type-I error of ARCHIE under the global null hypothesis of no association between the SNPs and gene network as shown in Fig. 2A.**

| Hypothesis | | Level | | | |
|---|---|---|---|---|---|
| | Method | $1 \times 10^{-03}$ | $1 \times 10^{-04}$ | $1 \times 10^{-05}$ | $1 \times 10^{-06}$ |
| At least one variant-gene association significant | Standard trans-eQTL | 0.15 | 0.044 | 0.004 | $2.1 \times 10^{-04}$ |
| At least one ARCHIE component significant | ARCHIE | $1.8 \times 10^{-04}$ | $9.1 \times 10^{-06}$ | $3.9 \times 10^{-07}$ | $2.5 \times 10^{-07}$ |

See Methods and Supplementary Section B for details on the simulation model.

of simulation iterations where there is at least one p-value across all possible variant-gene pairs is less than the chosen α-level. We found that ARCHIE maintains a highly conservative type-I error rate (Table 1) at all levels while standard trans-eQTL mapping maintains the overall familywise error rate of standard trans-eQTL mapping at 0.05 for the approximate Bonferroni correction level ($0.05/(40 \times 9) \approx 1 \times 10^{-04}$). Subsequently, we found that at approximately level $9 \times 10^{-04}$, ARCHIE achieves similar type-I error rate as the FWER of trans-eQTL mapping at level $1 \times 10^{-06}$. We use these levels for subsequent empirical power estimations.

We next compared the power of ARCHIE to identify gene-sets associated to trait-related SNPs in comparison to standard trans-eQTL mapping, for several different causal models (Fig. 2B, C; See Methods for details on the Simulation model). Briefly, each causal network consists of 40 SNPs associated to a trait, with successive five SNPs having regulatory effect on a local (cis) gene. In total eight cis (local) genes (in red) mediate the effect of these SNPs on a network of nine distal genes. The average cis heritability of the eight cis-genes were maintained approximately at 20–22% while the average trans-heritability for the nine distal genes were maintained at about 10%-14%. Each causal scenario was simulated at two sample sizes ($N = 1000$ and $30,000$) reflecting the approximate sample sizes of GTEx v8 and eQTLGen studies respectively.

For sparse causal networks (Fig. 2B), we found that the power of ARCHIE to detect genes in each layer of network was comparable or somewhat better than trans-eQTL mapping under two sample sizes (Fig. 2D). However, for the dense causal network (Fig. 2C), we find that ARCHIE has a major power advantage over trans-eQTL mapping throughout (Fig. 2E). This is potentially due to the increased weaker effects in the dense causal network (Fig. 2C) since the average trans-heritability is maintained at the same value as that of the sparse causal network. This demonstrates that ARCHIE can effectively aggregate weaker associations compared to trans-eQTL mapping. Additionally, we estimated power under the presence of a master regulator, where we found that as before ARCHIE had comparable or higher power that trans-eQTL of identifying the downstream genes (Supplementary Note 2 and Supplementary Fig. 2). We further compared the sensitivity and specificity of ARCHIE with that of standard trans-eQTL and found that although the sensitivity of ARCHIE was only slightly higher than, the specificity was substantially higher compared to trans-eQTL mapping (Supplementary Note 2 and Supplementary Fig. 3).

It is to be noted that the potential space of alternative null hypothesis for relationship between a group of SNPs and gene networks is extremely large, and a comprehensive evaluation is beyond the scope of this article. However, the above small scale simulations provide us with an intuitive insight that ARCHIE has robust power for detection of trans-association under a variety of plausible models for trans-associations.

We note that a primary objective of our analysis was to detect disease specific trans-association pattern in the background of broad trans-associations that are expected to be seen in the genome. Since GWAS variants associated to traits are enriched in

trans-eQTLs, we investigated whether ARCHIE captured any trait specific trans-association beyond the general expected pattern of trans-associations for GWAS variants which are not related to the trait in consideration. Thus, we considered this "*competitive null hypothesis*" to be more pertinent for our main analysis and we conducted additional resampling studies to investigate performance of ARCHIE for detecting trait-specific patterns. We performed resampling experiments using trans-eQTL summary statistics reported by eQTLGen consortium[19] across four different traits (See Sample description for details). For a given trait, we used summary statistics for 100 variants across 5000 genes of which a certain proportion ($\delta$) of the variants were related to the given trait and rest were variants randomly sampled from different traits reflecting the general background of trans-associations expected for GWAS variants (See Supplementary Note 2 for details). The results from applying ARCHIE, with varying $\delta$ (Supplementary Fig. 4), show that the probability of at least one ARCHIE component to be significant increases with the increase in the proportion of trait-specific variants ($\delta$), consistently across the four different traits. In particular, at $\delta = 0$, the results would correspond to the type-I error of ARCHIE under the competitive null hypothesis. We found that at a level of $1 \times 10^{-04}$, the competitive type-I error of ARCHIE is conservative for all the traits under consideration indicating that ARCHIE produces reduced false positives (Supplementary Data 1). Since competitive null hypothesis tests for trait-specific trans-association patterns, the subsequent results presented here are corresponding to the competitive null hypothesis unless otherwise mentioned.

The above numerical experiments taken together, demonstrate that ARCHIE can effectively identify weaker trait-specific trans-regulation effects.

**Trait-specific patterns of *trans*-associations in Whole Blood.** We applied ARCHIE on large-scale *trans*-association summary statistics for SNPs associated with 29 different traits reported by the eQTLGen consortium[19] to identify trans-regulated gene-sets associated with the respective traits. The eQTLGen consortium provides summary statistics ($Z$ values, $p$ values) from standard trans-eQTL mapping for more than 10,000 variants across numerous loci in the genome, curated from external databases, that have been identified to be associated to traits and diseases in large scale GWAS. For each of the 29 traits, we extracted the *trans*-association summary statistics for the variants associated with the trait and only the genes that were either more 5 Mb away from each of the variants or on a different chromosome. Thus, by design, all the genes in the analysis for a trait were distal to all the variants under consideration. We then applied ARCHIE to this *trans*-eQTL summary statistics and selected the trait-specific target genes via the significant gene components (See Methods and Fig. 1B for analysis details). On an average, across these traits, we detect 2 (max = 7 for "Height") significant sets of variant and gene components (ARCHIE components) capturing phenotype-specific *trans*-association patterns (Supplementary Fig. 1). Of the

**Table 2 The number of variants, genes and novel genes selected by ARCHIE for each the significant components for the analysis of SCZ, UC and PC.**

|  | Schizophrenia (SCZ) | Ulcerative Colitis (UC) |  | Prostate Cancer (PC) |  |
|---|---|---|---|---|---|
|  | Component 1 | Component 1 | Component 2 | Component 1 | Component 2 |
| Variants | 27 | 41 | 33 | 20 | 13 |
| Genes | 75 | 106 | 42 | 36 | 17 |
| Novel genes (proportion %) | 59 (78.7%) | 56 (52.8%) | 16 (38.1%) | 31 (86.1%) | 13 (76.5%) |

For a full list of selected variants and genes see Supplementary Data 2.

target genes selected by ARCHIE in the significant gene-components for each trait, approximately only 49.3% genes displayed a strong association in standard analysis (trans-eQTL $p$ value $< 1 \times 10^{-06}$ reported in eQTLGen) with any variant associated to that traits. The remaining 50.7% genes (termed "novel genes") harbors only weaker ($0.05 > p$ value $> 1 \times 10^{-06}$) associations and hence cannot be detected by standard trans-eQTL mapping alone; these genes display a similar pattern of trans-association with corresponding selected trait-related variants and are detectable only via the significant ARCHIE components. In fact, a large majority of the novel genes (89%) harbor multiple weaker trans-associations with the variants selected in the variant component. We made the list of target genes and variants selected by ARCHIE for each phenotype publicly available through an openly accessible database (https://github.com/diptavo/ARCHIE). Here, we focus on results for three different phenotypes, reflecting three different classes of diseases, their corresponding trans-association patterns, the selected target gene-sets, and the novel genes detected by ARCHIE (Table 2).

**Schizophrenia**. Schizophrenia is a neuropsychiatric disorder that affects perception and cognition. The eQTLGen consortium reports complete (non-missing) trans-association statistics for 218 SNPs, curated from multiple large-scale GWAS, associated with Schizophrenia (SCZ) across 7,756 genes. Of these, 7047 genes were expressed in whole blood of Genotype-Tissue Expression (GTEx)[28] v8 individuals. Testing against the competitive null hypothesis (See previous section), we identified one significant ARCHIE component capturing trans-association patterns significantly related to SCZ (Fig. 3A, B) consisting of 27 variants across several different genomic loci and 75 genes. Of the selected genes, only 16 (21.4%) had evidence of at least one strong association ($p$ value $< 1 \times 10^{-06}$) and possibly multiple weaker ($0.05 > p$ value $> 1 \times 10^{-06}$) association as reported by eQTLGen. The remaining 59 genes (78.6%) only had weaker trans-associations with SCZ-related variants (Table 2, Supplementary Data 2). These novel genes were not identified using traditional trans-eQTL mapping and were not reported as significant findings by the eQTLGen consortium. We further investigated the robustness of the results using coexpression estimates from gene expressions levels reported in Whole Blood by the Depression Genes and network (DGN) study[29]. The results show that out of 75 genes and 59 novel genes identified by ARCHIE, 62 and 45 were also selected in the replication analysis using data from DGN, demonstrating the potential robustness of the results by ARCHIE.

Next, we tested whether the selected variants and genes were enriched in trans-heritability for SCZ. Using an expression imputation approach (See Methods for details), we estimated the approximate heritability of SCZ explained by the trans-associations between the selected genes and variants, using individual level data from UK Biobank. We compared the estimate to (1) the expected distribution of heritability explained by the selected SNPs and randomly chosen genes (excluding the

selected genes) for SCZ and (2) the expected distribution of heritability explained by the selected SNPs and genes for a randomly chosen trait. The results showed that the selected SNPs and genes are significantly enriched in trait heritability ($p$ value $< 0.001$) than expected by chance (Fig. 3D).

Several of the 59 identified novel genes have previously been reported to be associated with neurological functions. For example, chemokine receptor 4 (CXCR4), a gene that underlies interneuron migration and several neurodegenerative diseases[30], was identified by aggregating weaker associations from 20 SCZ-related SNPs in the variant component, but does not have any significant trans-associations. Similarly, caveolin-1 (CAV1), which is a known regulator of a SCZ risk gene (DISC1)[31], aggregates 13 weaker association to SCZ-related variants in the variant component. Notably, the target genes identified by ARCHIE include genes such as HSPA5 and AP5S1, which not only harbor multiple trans-associations from SCZ-related variants but have also been reported to have cis-variants associated with psychiatric disorders[32,33]. We investigated whether in general the genes selected by ARCHIE had have evidence of association with SCZ through cis variants. Aggregating results from several large-scale cis-eQTL studies across tissues[9,34], we found that 12 of the 59 of the (enrichment $p$ value $= 2.8 \times 10^{-05}$) novel genes have nominally significant ($p$ value $< 1 \times 10^{-04}$) evidence of cis-regulatory SNPs to be associated with SCZ or other different neuropsychiatric diseases. Further, eQTLGen consortium reports the association of gene expressions with several publicly available polygenic risk scores (PRS) of SCZ at different $p$ value thresholds, curated from external databases. We found that out of the 59 novel genes identified, 38 (64.4%) had a nominal association with at least one reported PRS for SCZ.

By performing pathway enrichment analysis of the target genes using FUMA[35] (https://fuma.ctglab.nl) and ShinyGO[36] (http://bioinformatics.sdstate.edu/go), we investigated if the selected genes represented known SCZ-related biological mechanisms (See Methods for details). Among the significantly enriched pathways, the majority (51.3%) were immune related. In particular, we identified 42 GO pathways[37], 36 canonical pathways[38–41] and 4 hallmark pathways[42] to be strongly enriched (FDR adjusted $p$ value $< 0.05$) for the selected genes with 73 (89.0%) of them containing at least one novel gene (Fig. 3C, Table 3 and Supplementary Data 3). Several pathways, previously reported in connection to SCZ, are identified to be enriched (FDR adjusted $p$ value $< 0.05$) for the selected genes (Fig. 3C). For example, among the enriched gene ontology (GO) terms, GO-0034976: response to endoplasmic reticulum stress[43] (FDR adjusted $p$ value $= 0.013$), GO-055065 metal ion homeostasis[44] (adjusted $p$ value $= 0.029$), GO-0006915: apoptotic process[45] (adjusted $p$ value $= 0.029$), GO-0043005: neuron projection (adjusted $p$ value $= 0.021$) have previously been suggested to be linked to SCZ (Table 3). Four hallmark gene-sets are also found to be significantly enriched for the selected genes including glycolysis[46], hypoxia[47], mTORC1 signaling[48], and unfolded protein response[49], all of which have suggestive evidence of being

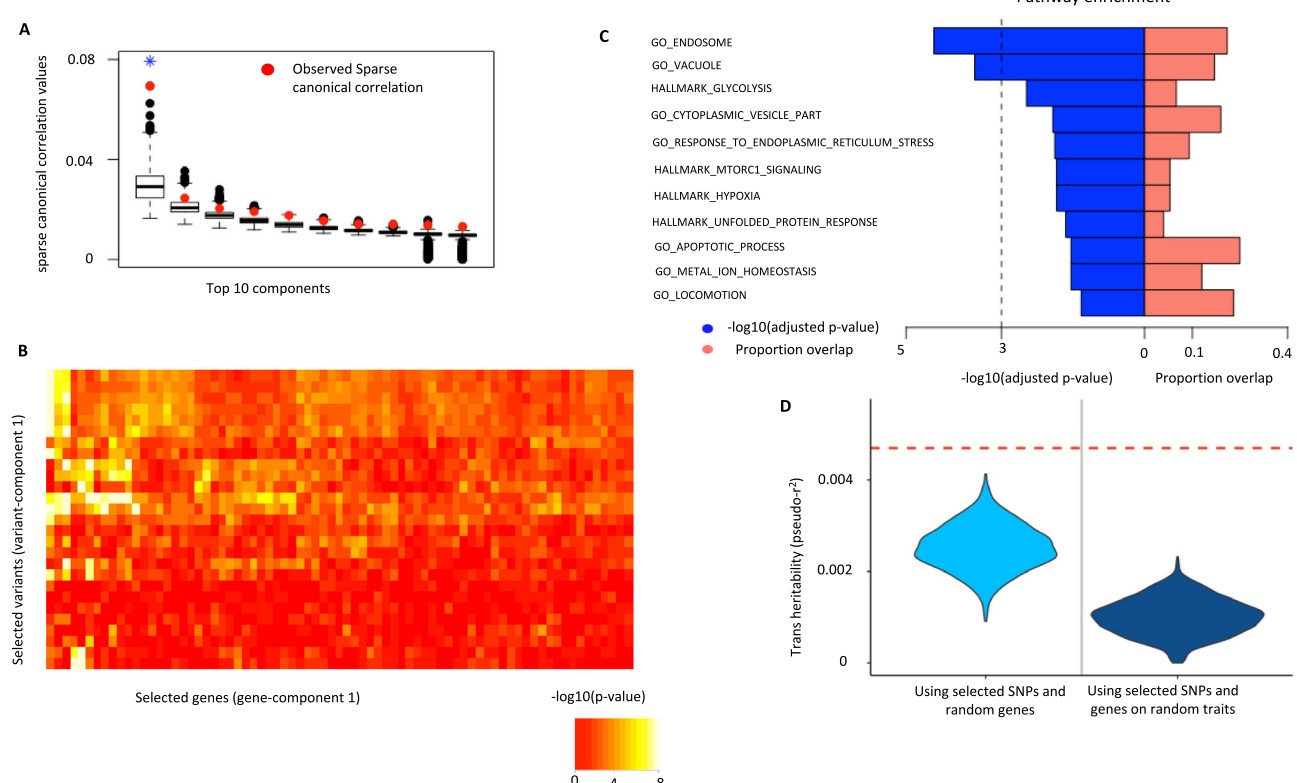

**Fig. 3 Trans-association pattern and properties of selected target gene-set for Schizophrenia (SCZ). A** Top 10 sparse canonical correlation values (cc-values; red) and the corresponding *competitive* null distributions (black box plot) for ARCHIE analysis of SCZ. Top 1 ARCHIE component is significant. The box plot for competitive null distribution is presented as ±1.5 interquartile range with the median line denoted at the center. The competitive null distribution denotes the general pattern of sparse canonical correlation (and trans-associations) expected in analysis of variants that are not specific to SCZ. The significant ARCHIE components are marked by a blue asterisk. **B** $-\log_{10}$ (p values) for standard trans-eQTL association test between variants and genes selected in ARCHIE component 1 as reported in eQTLGen (two sided test). Any association p value $< 10^{-08}$ is collapsed to $10^{-08}$ for the ease of viewing. **C** Pathway enrichment results for some selected top pathways for the target genes selected in gene-component 1. For full pathway enrichment results for the target genes selected see Supplementary Data 3, 4. The dashed vertical line corresponds to a suggested FDR threshold of 0.001. **D** trans-heritability enrichment analysis for target genes selected for SCZ. The violin plots represent the distribution of pseudo-$r^2$ for the selected SNPs and a random gene-set of the same size on SCZ (left) and the distribution of pseudo-$r^2$ of the selected genes and SNPs on a random trait in UK Biobank (right). The red dashed line represents the observed pseudo-$r^2$ for the selected SNPs and target genes for SCZ (See Results and Methods for details).

**Table 3 Pathway enrichment results for the target genes selected for SCZ.**

| Category | Pathway | Adjusted p value | Genes in pathway | Genes overlap | Novel genes overlap |
|---|---|---|---|---|---|
| GO | Endosome (GO: 0005768) | $3.8 \times 10^{-05}$ | 885 | 13 | 11 |
| | Vacuole (GO: 0005773) | $2.7 \times 10^{-04}$ | 760 | 11 | 9 |
| | Response to endoplasmic reticulum stress (GO: 0034976) | $1.3 \times 10^{-02}$ | 272 | 7 | 6 |
| Hallmark | Glycolysis | $3.3 \times 10^{-03}$ | 200 | 5 | 4 |
| | Hypoxia | $1.4 \times 10^{-02}$ | 200 | 4 | 3 |
| | MTORC1 signaling | $1.4 \times 10^{-02}$ | 200 | 4 | 3 |
| Curated Gene sets | Wierenga_STAT5A_TARGETS_DN | $5.9 \times 10^{-03}$ | 189 | 6 | 5 |
| | Zhou_INFLAMMATORY_RESPONSE_LIVE_DN | $6.5 \times 10^{-03}$ | 372 | 7 | 3 |
| Oncogenic | TBK1. DN.48HRS_DN | $3.1 \times 10^{-02}$ | 50 | 3 | 3 |
| Signatures | ATF2_S_UP. V1_DN | $4.4 \times 10^{-02}$ | 185 | 4 | 4 |
| Immunologic | GSE6269_E_COLI_VS_STREP_PNEUMO_INF_PBMC_DN | $3.9 \times 10^{-07}$ | 166 | 9 | 8 |
| Signatures | GSE6269_E_COLI_VS_STAPH_AUREUS_INF_PBMC_DN | $2.1 \times 10^{-04}$ | 174 | 7 | 5 |

Several selected top pathways containing novel genes across different categories are shown. See Supplementary Data 3 and 4 for results on all significant pathways and transcription factor target gene-sets for the selected genes.

associated to SCZ. Using numerous TF databases[50,51], we found that the selected target genes were enriched (adjusted p value < 0.05) for targets of ten TFs (Supplementary Data 4), several of which have been previously reported to be associated with

neuropsychiatric disorders[52,53]. While connection between immune pathways and SCZ is less obvious, we believe this is also a noteworthy finding given recent studies have increasingly pointed out complex interactions between the immune system,

inflammation, and the brain[54]. However, relative importance of different pathways is likely to have been biased in our analysis because blood is more likely to contain stronger signature of immune mechanisms than processes that are directly related to brain.

Protein-protein interaction (PPI) enrichment analysis using STRING (v11.0)[55] showed a significant enrichment ($p$ value = 1.1 × 10$^{-03}$) indicating that the corresponding proteins may physically interact. Although the analysis is performed using trans-associations in whole blood, we found several of the selected target genes were differentially expressed in tissues directly related with SCZ. For example, three novel genes (*PADI2, KCNJ10, MLC1*), were highly upregulated in several brain tissues (Supplementary Fig. 5), in comparison to their expression in rest of the tissues in GTEx v8. To systematically evaluate this, we performed a differential expression enrichment analysis to investigate whether the target genes were differentially expressed in any of the 54 tissues in GTEx v8 dataset. Since SCZ is a complex disease, we expect the selected genes to be differentially expressed or regulated in numerous potentially distinct tissues. For each tissue, we curated lists of differentially expressed genes across the genome. We defined a gene to be differentially expressed in a tissue if the corresponding gene expression level in that tissue was significantly different from that across the rest of the tissues (See Supplementary Note 3 for details). Using such pre-computed lists of differentially expressed genes for each tissue, we found that the target genes selected by ARCHIE were enriched within the set of differentially expressed genes in 12 different tissues including 7 brain tissues in

GTEx v8 (Supplementary Fig. 6). Majority of these tissues had been previously reported to be involved in the pathophysiology of SCZ[56–58].

**Ulcerative colitis**. Ulcerative colitis (UC) is a form of inflammatory bowel disease, affecting the innermost lining of colon and rectum, causing inflammation and sores in the digestive tract and can lead to several colon-related symptoms and complications including colon cancer[59–61]. The eQTLGen consortium reports complete (non-missing) *trans*-association summary statistics for 163 SNPs associated with Ulcerative Colitis, curated from multiple large-scale GWAS, across 12,010 genes. Of these, 10,307 genes were expressed in Whole Blood from GTEx v8 individuals. Using ARCHIE, we detected two significant variant-gene components comprising of 74 SNPs and 148 genes in total (Fig. 4A and Supplementary Fig. 7; Supplementary Data 2) that reflect *trans*-association patterns specific to UC. Of the selected genes, 68 genes (45.9%) were novel, meaning they did not have any strong *trans*-association (Table 2, Supplementary Data 2) with the variants related to UC. Further, similar to SCZ, we found the associations of the SNPs with target genes was strongly enriched ($p$ value < 0.001) for heritability of UC than expected by chance alone and also the selected trans-associations explained heritability of UC more than expected for a random trait or disease (Fig. 4D).

Several of the novel target genes detected have been previously linked to intestinal inflammations and diseases. For example,

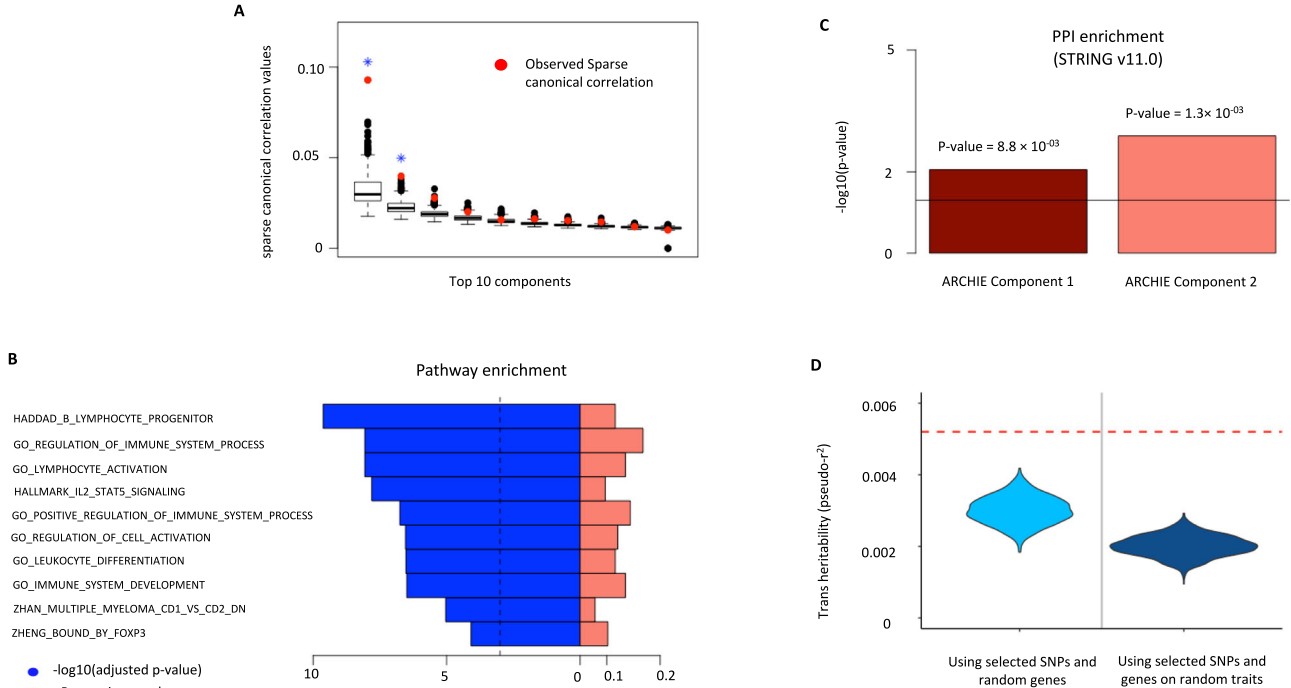

**Fig. 4 Analysis of selected target gene-set for Ulcerative Colitis (UC). A** Top 10 cc-values (red) and the corresponding *competitive* null distributions (black box plot) for ARCHIE analysis of UC. Top 2 ARCHIE components are significant. The box plot for competitive null distribution is presented as ±1.5 interquartile range with the median line denoted at the center. The competitive null distribution denotes the general pattern of sparse canonical correlation (and trans-associations) expected in analysis of variants that are not specific to UC. The significant ARCHIE components are marked by a blue asterisk. **B** Pathway enrichment results (one sided test) for some selected top pathways for the selected target genes. The dashed vertical line corresponds to a suggested FDR threshold of 0.001. For full pathway enrichment results for the target genes selected for both the components see Supplementary Data 5–7. **C** PPI enrichment (one sided test) for the two significant ARCHIE components for UC. **D** trans-heritability enrichment analysis for target genes selected for UC. The violin plots represent the distribution of pseudo-r$^2$ for the selected SNPs and a random gene-set of the same size on UC (left) and the distribution of pseudo-r$^2$ of the selected genes and SNPs on a random trait in UK Biobank (right). The red dashed line represents the observed pseudo-r$^2$ for the selected SNPs and target genes for UC (See Results and Methods for details).

glycoprotein A33 (*GPA33*) is known to impact intestinal permeability[62] and is an established colon cancer antigen[63]. Recent research using mouse models have reported a connection between the regulation of *GPA33* and the development of colitis and other colon-related inflammatory syndromes[64]. We also identify spermine oxidase (*SMOX*) through its weaker association with 9 UC-related variants. *SMOX* is significantly upregulated in individuals with inflammatory bowel diseases[65] and has been implicated in gastric and colon inflammations as well as carcinogenesis[66].

Using a series of follow-up analyses, we identify several pathways to be enriched (FDR adjusted *p* value < 0.05) for the selected target genes (Supplementary Data 5, 6), majority of them being immune related (59.6%). Among others, the hallmark interleukin-2-STAT5 signaling pathway (FDR adjusted *p* value = $1.6 \times 10^{-08}$) has previously been reported to be associated to development of UC via suppression of immune response[67]. Various GO pathways related to endocytosis, lymphocyte activation, T-cell activation are found to be overrepresented in the selected target genes as well (Fig. 4B and Table 4). Further enrichment analysis using broad TF databases, we found the selected target genes across both gene components are enriched (adjusted *p* value < 0.01) for targets of 18 different TFs, majority of which have been previously reported to be involved in mucosal inflammation, inflammation of the intestine and epithelial cells and in immune-related responses (Supplementary Data 7).

PPI enrichment analysis shows that the resultant proteins interact more often than random (*p* value = $8.8 \times 10^{-03}$ and $1.3 \times 10^{-03}$ respectively for two significant ARCHIE components) (Fig. 4C). Additionally, the selected genes were found to be enriched for genes significantly differentially expressed in several relevant tissues like colon-sigmoid and small-intestine ileum among others (Supplementary Fig. 8).

We further investigated if any known mechanism can explain how the selected genes are associated with the selected variants, including mechanisms reflecting *cis* mediation[15]. In one example from our analysis, we observe that, among the 41 variants selected by variant-component 1, one UC-related variant rs3774959 is a *cis*-eQTL of *NFKB1* (*p* value = $6.2 \times 10^{-41}$ in eQTLGen and $6.3 \times 10^{-05}$ in GTEx in whole blood). The Nuclear factor κB (NF-κB) family of transcription factors (TF) including *NFKB1*, has been extensively reported to be involved in immune[68] and

inflammatory responses[69]. In particular, mutations in the promoter region of Nuclear factor κB1 (*NFKB1*) have been strongly implicated to be associated with UC[70], although the downstream target genes of *NFKB1* that are associated with UC, are largely unknown. Among 106 target genes selected in the first gene component, there are 6 genes (*CD74, CD83, IL1B, Il2RA, PTPN6, FOXP3*) that are reported targets for *NFKB1* (adjusted enrichment *p* value = $7.5 \times 10^{-03}$) in TRRUST v2.0[50]. Thus, it can be conceptualized that the selected UC-related variant may regulate the expression levels of the six selected targets of *NFKB1* via *cis*-regulation of *NFKB1* expression levels, influencing UC-status downstream.

**Prostate cancer.** Prostate cancer (PC) is one of the most common types of cancers in middle-aged and older men, having a high public health burden with more than 3 million new cases in USA per year. The eQTLGen consortium reports complete (non-missing) *trans*-association summary statistics for 122 SNPs associated with prostate cancer, curated from multiple large-scale GWAS, across 12,951 genes. Of these, 11,385 genes were expressed in Whole Blood from GTEx v8 individuals. Using ARCHIE, we detected two significant variant-gene components comprising 33 SNPs, spanning 14 different loci across the genome, and 53 genes in total (Fig. 5A; Supplementary Data 2) that reflect *trans*-association patterns specific to PC, of which 44 genes (83.1%) were novel (Table 2, Supplementary Data 2). Additionally, similar to SCZ and UC, we found evidence of enrichment of trans-heritability of PC that can be mediated by the target genes and also that the trans-heritability mediated by the selected SNPs and genes was significantly more than that for a randomly chosen trait (Fig. 5D), but the level of significance achieved was relatively weaker (*p* value = 0.001 and 0.007; See Methods for details).

Among the novel genes, we identified several key genes that are generally implicated in different types of cancers. For example, *TP53* aggregates weaker *trans*-associations with 9 PC-related variants in ARCHIE component 1 (Fig. 5B). The *TP53* gene encodes tumor protein p53 which acts as a key tumor suppressor and regulates cell division in general. *TP53* is implicated in a large spectrum of cancer phenotypes and has been considered to be one of the most important cancer genes studied[71]. Further, genes associated with the second gene component included *SMAD3*

**Table 4 Pathway enrichment results for the target genes selected for UC.**

| Component | Category | Pathway | Adjusted *p* value | Genes in pathway | Genes overlap | Novel genes overlap |
|---|---|---|---|---|---|---|
| 1 | GO | Regulation of Immune system (GO: 0002683) | $8.5 \times 10^{-09}$ | 1606 | 25 | 11 |
| | | Lymphocyte activation (GO: 0046649) | $8.5 \times 10^{-09}$ | 709 | 18 | 8 |
| | | Regulation of T-cell activation (GO: 0050863) | $2.9 \times 10^{-06}$ | 311 | 11 | 6 |
| | Hallmark | IL2 STAT5 signaling | $3.2 \times 10^{-10}$ | 200 | 10 | 5 |
| | | G2M checkpoint | $2.6 \times 10^{-03}$ | 200 | 4 | 4 |
| | Curated | PILON_KLF1_TARGETS_DN | $2.5 \times 10^{-05}$ | 854 | 11 | 4 |
| | | KEGG: Endocytosis | $3.2 \times 10^{-02}$ | 181 | 5 | 3 |
| | | Reactome: Adaptive immune system | $2.1 \times 10^{-02}$ | 807 | 10 | 3 |
| | Immunologic | GSE7460_WT_VS_FOXP3_HET_ACT_WITH_TGFB_TCONV_UP | $6.9 \times 10^{-08}$ | 200 | 10 | 8 |
| | Signatures | GSE4984_UNTREATED_VS_GALECTIN1_TREATED_DC_DN | $4.5 \times 10^{-16}$ | 191 | 16 | 7 |
| | | GSE22886_NAIVE_CD4_TCELL_VS_MONOCYTE_UP | $3.2 \times 10^{-07}$ | 196 | 11 | 6 |
| 2 | GO | Defense Response (GO: 0006952) | $2.0 \times 10^{-04}$ | 1684 | 11 | 4 |
| | | Inflammatory Response (GO: 0022610) | $1.2 \times 10^{-03}$ | 714 | 7 | 2 |
| | Curated | JAATINEN_HEMATOPOIETIC_STEM_CELL_DN | $4.7 \times 10^{-04}$ | 233 | 6 | 3 |
| | Immunologic | GSE22886_NAIVE_BCELL_VS_MONOCYTE_UP | $2.1 \times 10^{-12}$ | 195 | 11 | 4 |
| | Signatures | GSE10325_CD4_TCELL_VS_BCELL_DN | $5.8 \times 10^{-08}$ | 193 | 8 | 3 |

Several selected top pathways containing novel genes across different categories are shown here. See Supplementary Data 5–7 for results on all the significant pathways for the selected in ARCHIE components 1 and 2, respectively.

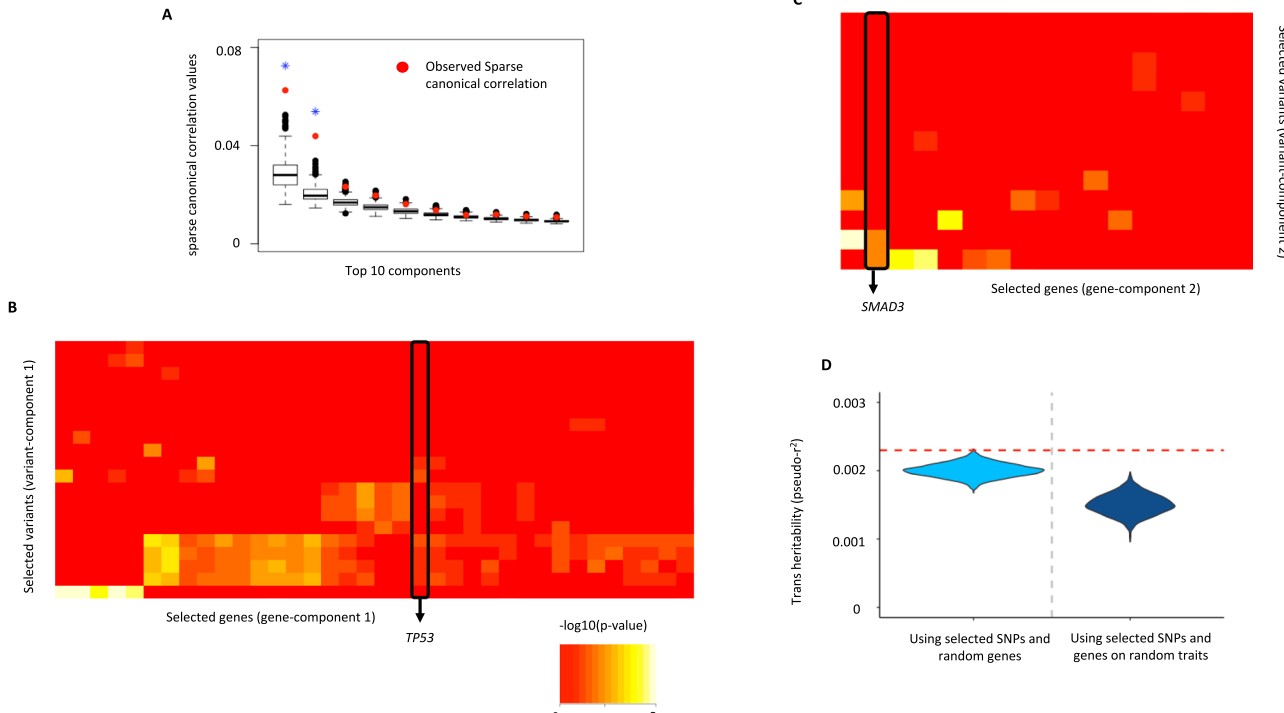

**Fig. 5 Analysis of selected target gene-set for Prostate Cancer (PC). A** Top 10 cc-values (red) and the corresponding competitive null distributions (black box plot) for ARCHIE analysis of PC. Top 2 ARCHIE components are significant. The box plot for competitive null distribution is presented as ±1.5 interquartile range with the median line denoted at the center. The competitive null distribution denotes the general pattern of sparse canonical correlation (and trans-associations) expected in analysis of variants that are not specific to PC. The significant ARCHIE components are marked by a blue asterisk. -$\log_{10}$ p-value for trans-eQTL association between variants and genes selected in **B** ARCHIE component 1 and **C** component 2 as reported in eQTLGen data. *TP53* and *SMAD3* are highlighted. Any association $p$ value $< 10^{-08}$ is collapsed to $10^{-08}$ for the ease of viewing. **D** trans-heritability enrichment analysis for target genes selected for PC. The violin plots represent the distribution of pseudo-$r^2$ for the selected SNPs and a random gene-set of the same size on PC (left) and the distribution of pseudo-$r^2$ of the selected genes and SNPs on a random trait in UK Biobank (right). The red dashed line represents the observed pseudo-$r^2$ for the selected SNPs and target genes for PC (See Results and Methods for details).

(Fig. 5C) which is also a well-known tumor suppressor gene that plays a key role in transforming growth factor β (TGF-β) mediated immune suppression and also in regulating transcriptional responses suitable for metastasis[72–74]. *TP53* and *SMAD3* belong to two different ARCHIE components meaning that they might pertain to two relatively distinct biological processes that are independently affected by different sets of PC-related variants. Additionally, the second gene component included *EEA1* which is reported to have significantly altered expression levels in prostate cancer patients[75].

Using enrichment analyses (Supplementary Data 8, 9), we found several pathways, including broadly ubiquitous pathways to be significantly overrepresented in the selected genes for both the gene components like regulation of intracellular transport (adjusted $p$ value = 0.017) and mRNA 3'-UTR binding (adjusted $p$ value = 0.008). Notably, we found the selected genes to be enriched for targets of several transcription factors many of which have been associated with different types and subtypes of cancer (Supplementary Data 10). For example, we found a TF target enrichment for *SPAG9* (adjusted p-value = 0.016) which has been identified to be associated with breast cancer, ovarian cancer, colorectal cancer and others[76]. We also found enrichment for targets of *SSRP1* (adjusted $p$ value = 0.016), which is differentially regulated in a wide spectrum of malignant tumors[77] along with enrichment for targets of *MYC* and *TP73* which are well-established cancer-related genes (Supplementary Data 10). However, we did not find any evidence for significant enrichment of PPI among identified genes.

The downstream analysis suggests that a majority of the pathways (78.1%) enriched for the selected genes are immune related as observed in the previous examples as well. This might have been driven by the fact that eQTLGen reports summary *trans*-associations in whole blood. In general, whole blood might not be the ideal candidate tissue to identify *trans*-associations pertaining to PC. It is conceivable that relevant tissue-specific analysis for PC could have illuminated further *trans*-association patterns and identified key tissue-specific target genes. Despite that, we can identify several genes which have been elaborately reported to be key target genes for various cancers as well as some novel *trans*-associations. This underlines the utility of our aggregative approach and that it can illuminate important target genes pertaining to a trait.

## Discussion

While modern genome-wide association studies have been successful in identifying a large number of genetic variants associated with complex traits, the underlying biological mechanisms by which these associations arise has remained elusive. Although trans genetic regulations, mediated through cis- or otherwise, have been proposed for detecting important target genes for GWAS variants, identification of trans associations using standard univariate SNP vs gene-expression association analysis is notoriously difficult due to weak effect-sizes and large multiple testing burdens. In this article, we have proposed ARCHIE, a summary statistics-based method for identifying groups of trait-associated genetic variants whose effects may be mediated

through *trans*-associations with groups of coregulated genes. ARCHIE involves estimating LD and coexpression from publicly available reference datasets. However, we found that the results can be reasonably robust to the choice of reference data and estimation errors arising due to sampling variations (Supplementary Fig. 9). In general, robust performance can be achieved under an optimal ratio of the sample sizes of the reference transcriptomic study and the study from which summary statistics of standard *trans*-eQTL associations are being analyzed. Further, we develop a resampling-based method to test the statistical significance of trait-specific enrichment patterns in the background of expected highly polygenic broad trans association signals. We have shown through simulation studies that compared to standard *trans*-eQTL analysis, ARCHIE is more powerful for the detection of "core"-like genes which may potentially mediate the effects of multiple upstream genes and variants and can explain trait-specific genetic associations.

Application of the method to eQTLGen consortium *trans*-eQTL statistics not only identified many novel *trans*-associations for trait-related variants, but also helped to contextualize the individual associations in terms of broader trait-specific trans-regulation patterns that were detected by underlying gene and variant components. The set of selected target genes in the gene component is one of the key outputs of ARCHIE. Using a series of follow-up analyses for three different types of traits, we showed that the selected genes are often overrepresented in known disease-relevant pathways, enriched in protein-protein interaction networks, show co-regulations across tissues and contain targets for known transcription factors implicated in the disease (SCZ and UC) and key tumor suppressor genes (PC). Further, using a *trans*-expression imputation approach, we demonstrated that the selected genes can significantly mediate heritability-associated trait-related variants. All of these analyses point out that the *trans*-association patterns we detect are likely to have trait-specific biological basis.

There are several limitations of the proposed method and current analysis. First, in the current version ARCHIE, we begin with a set of genetic variants associated with a trait, but we do not incorporate the underlying association directions and effect sizes in the analysis. This approach allowed us to independently investigate identified target genes through testing for consistency of directions of association of the SNPs with the trait and those with the expressions of the target through the *trans*-heritability analysis. However, it is likely that incorporation of the GWAS effect sizes (value and direction) of trait association for the SNPs in the sCCA itself will lead to improved power for detection of the trait-specific target genes. One of the immediate ways to incorporating that might be to weight the variants with weights proportional to squared (or absolute value) of the GWAS effect size. However, incorporating the direction of the GWAS effect with the sCCA framework remains an interesting problem and merits further research. Additionally, incorporation of information on cis-genes and known functional annotation of genetic variants can improve the power of the analysis as well.

Although the estimation of the ARCHIE components is computationally efficient, in the current implementation, the resampling-based testing method is computationally intensive. In the future, further research is merited to develop analytical approximation techniques to reduce the computational burden of ARCHIE. Since variable selection is a major goal in ARCHIE, we have introduced regularization via the L1 penalty due to its proven theoretical selection consistency[78]. In fact, in presence of correlation between SNPs as well gene expressions, elastic net regularization approach can be effective in accurate selection. In the future, usefulness of alternative types of penalty functions in selecting the genes and variants merits further research. Further, due to the lack of existing methods to identify trait-specific trans associations, we

have compared the performance against the standard trans-eQTL mapping. Although the eQTLGen data analysis shows that ARCHIE can identify a broader range of gene-sets trans-regulated by GWAS variants as compared to standard trans-eQTL mapping, the goals of ARCHIE and the standard analysis are different and therefore not directly comparable—the goal of the standard analysis is to identify association in individual variant-gene pair irrespective of trait specificity. In contrast, ARCHIE can identify trait-specific trans-regulated target genes harboring multiple weaker associations with trait-related variants. As newer methods are developed to detect trait-relevant trans-regulated genes, more comprehensive analyses comparing alternative approaches will allow us to understand their benefits and limitations.

Currently, not many transcriptomic studies have made the summary statistic from trans-eQTL mapping available. However, as these studies grow in size and with improved methods of data sharing, we expect more studies to make the summary statistics from trans-eQTL mapping available, allowing researchers to investigate a broader range of diseases and traits. Further, ARCHIE can be broadly applicable to understand role of other types of molecular traits, such as proteins and metabolites, in mediating complex trait genetic associations. In the future, as data on molecular biomarkers become increasingly available in large biobanks, tools like ARCHIE will be increasingly needed to understand common pathways through which genes and biomarkers interact to cause specific diseases.

In this article, we have analyzed summary statistics reported by eQTLGen in the whole blood. This is primarily because of the substantial effective sample size of eQTLGen. While the approach can be applied to eQTL results from other tissues, the underlying sample sizes may be too limited to yield sufficient power. Although blood might not be the most relevant tissue for a number of traits, our analyses did detect trans association patterns that appear to have a broader biological basis in the disease genetics, from multiple independent lines of evidence. Nevertheless, it is likely that our analysis has missed many *trans*-association patterns that will be present only in specific disease-relevant tissues, cell types or/and dynamic stages[79]. In the future, we will seek applications for ARCHIE in various types of emerging eQTL databases to provide a more complete map of networks of genetic variants and *trans*-regulated gene expressions and relevant contexts.

In summary, in this article we have developed a summary-based method, ARCHIE, to detect trait-specific gene-sets by aggregating *trans*-associations from multiple trait-related variants. ARCHIE is a powerful tool for identifying target gene sets through which the effect of genetic variants on a complex trait may be mediated. In the future, applications of the methods to a variety of existing and new data on association between genetic variants with high-throughput molecular traits can provide insights to biological mechanisms underlying genetic basis of complex traits.

## Methods

**Sample Description**. eQTLGen: The eQTLGen consortium[19] is a large-scale multi-study effort to identify to study the downstream effects of trait-related variants via their effects on gene expression in whole blood. The consortium consists of 37 individual studies with a collective sample size of 31,684 participants. With this sample size, the study has relatively higher power to detect moderate to weaker effects of variants on gene expression. 10,317 variants related to complex traits, compiled from several GWAS databases, were tested for *trans*-associations with the expression levels of 19,964 genes in whole blood. The authors have made summary statistics ($Z$ score, $p$ value) for these *trans*-eQTL mapping analyses freely available to public.

GTEx: The Genotype-Tissue Expression (GTEx) project[28] aims to study tissue-specific gene expression and regulation. We used individual-level data from GTEx (v8) whole blood to construct the co-expression matrix ($\Sigma_{EE}$) and further downstream validation of the gene-sets selected using ARCHIE. In our analysis, we

used the latest version (v8) of GTEx having gene expression and genotype data with samples from 54 different tissues. In particular, 755 individuals had expression data on 20,315 genes for whole blood. Of these, we used 670 individuals with genotype data present.

UK Biobank: UK Biobank is a large biobank study with above 500,000 participants. Among several data resources available, the genotype data, hospitalization records, and health-records data are available. We used individual-level genotype data from UK Biobank to construct LD matrix ($\Sigma_{GG}$) and for further downstream analysis of the selected target genes.

The phenotype data constructed from hospitalization and health-data records were used in the quantification and testing of enrichment in *trans*-heritability explained by the selected target genes (See Methods). We included the individuals with European ancestry in the analysis. For example, in the analysis of SCZ, we used a sample of 366,326 participants from UK Biobank to construct the imputed gene-expression levels and evaluate the corresponding regression $r^2$ as an estimate of *trans*-heritability on SCZ as a binary phenotype.

**Estimating trait-specific pattern of *trans*-associations.** Our proposed method, Aggregative trans association to detect phenotype-specific gene-sets (ARCHIE), can select target genes trans-associated with trait-related variants using summary statistics in a sparse canonical correlation framework. To introduce the details of ARCHIE, first, we briefly describe sCCA. For n individuals, let $G^{n \times p}$ be the normalized genotype matrix for $p$ variants and $E^{n \times g}$ be the normalized gene-expression matrix for g genes all of which are distant (trans) to the variants. sCCA seeks to estimate sparse linear combinations of variants ($u^{p \times 1}$) and genes ($v^{g \times 1}$) such that the correlation between Gu and Ev is maximized i.e.,

$$(\mathbf{u}, \mathbf{v}) = \mathbf{argmax} \ \bar{\mathbf{v}}^T \mathbf{E}^T \mathbf{G} \bar{\mathbf{u}}$$
$$\text{with } \mathbf{v}^T \mathbf{E}^T \mathbf{E} \mathbf{v} \leq 1 \text{ and } \mathbf{u}^T \mathbf{G}^T \mathbf{G} \mathbf{u} \leq 1 \text{ and } \|\tilde{\mathbf{U}}\|_1 \leq c_u; \|\tilde{\mathbf{V}}\|_1 \leq c_v \quad (1)$$

where $||\mathbf{x}||_h$ is the $L_h$ norm of a vector x; $c_u$ (or $c_v$) is the sparsity parameter on the variant (or gene) component for the lasso-type $L_1$ penalty. The subsequent pairs of sCCA components are obtained by similarly maximizing the correlation between **Gu** and **Ev** and under the constraint of being uncorrelated or orthogonal to the previous components. In practice, the optimization problem can be reformulated in terms of covariance matrices as:

$$(\mathbf{u}, \mathbf{v}) = \mathbf{argmax} \ \bar{\mathbf{v}}^T \mathbf{W} \bar{\mathbf{u}}$$
$$\text{with } \|\bar{\mathbf{u}}\|_1 \leq c_u; \|\bar{\mathbf{v}}\|_1 \leq c_v \text{ and } \|\bar{\mathbf{u}}\|_2 = 1, \|\bar{\mathbf{v}}\|_2 = 1 \quad (2)$$

where $W = \Sigma_{EE}^{-1/2} \Sigma_{GE} \Sigma_{GG}^{-1/2}$ and $v_k = Wu$; $\Sigma_{GG}$ and $\Sigma_{EE}$ are the column-correlations of **G** and **E** respectively and $\Sigma_{GE}$ is the cross-covariance matrix between the variants and the gene expressions. However, the matrices can be estimated from publicly available summary statistics and reference data. We approximate $\Sigma_{GG}$ and $\Sigma_{EE}$ by the empirical LD-matrix and a penalized co-expression matrix (See Supplementary Note 1) using external reference samples and $\Sigma_{GE}$ can be obtained from the summary statistics of the regression for trans-eQTL mapping across all pairs of variants and gene-expressions (See Supplementary Note 1).

To apply ARCHIE, thus, we start with the summary statistics from *trans*-eQTL mapping (Z value, p value). Given the *trans*-association summary statistics across the variants related with the trait and all the corresponding distant genes (variant > 5 Mb away from the transcription start site of the gene), we first adjust for the correlation within the variants and genes through appropriate LD and co-expression matrices, estimated from reference panels with individual-level genotype (or dosage) and gene expression data respectively, as follows:

$$W = \Sigma_{EE}^{-1/2} \Sigma_{GE} \Sigma_{GG}^{-1/2} \quad (3)$$

where $\Sigma_{GG}$ and $\Sigma_{EE}$ are estimates of LD-matrix and co-expression matrix (see Supplementary Note 1), and $\Sigma_{GE}$ is the cross-correlation matrix obtained using the Z values from the standard trans-eQTL mapping across all pairs of variants and gene-expressions. It is important to adjust for the dependence within the variants and gene expression levels using the LD and co-expression matrices respectively, since we aim to identify trait-relevant gene-sets and variants through independent trans-associations. Furthermore, gene-expression levels in bulk tissues may appear to be correlated due to cell composition effects as well, which needs to be adjusted for by incorporating the estimated co-expression between the genes.

Using W, the correlation-adjusted matrix of trans-associations, ARCHIE employs sCCA[22,24] which produces a sparse linear combination of the variants (u; termed variant-component) that is strongly correlated with a sparse linear combination of genes (v; termed gene-component) by solving the following optimization problem

$$(\mathbf{u}, \mathbf{v}) = \arg \max \ \bar{\mathbf{v}}^T \mathbf{W} \bar{\mathbf{u}}$$
$$\text{with } \|\bar{\mathbf{u}}\|_1 \leq c_u; \|\bar{\mathbf{v}}\|_1 \leq c_v \text{ and } \|\bar{\mathbf{u}}\|_2 = 1, \|\bar{\mathbf{v}}\|_2 = 1 \quad (4)$$

Sparsity aids in interpretation since each non-zero element of a variant or gene component indicates that the respective variant (or gene) is selected in that component. Thus, (u v), which are the resultant variant and gene components (jointly termed ARCHIE components) can be interpreted as the sparse latent factors that explain the majority of the aggregated association between all the trait-related variants and all the genes. The corresponding sparse canonical correlation (cc-value) between each pair of

variant and gene components, defined as $q^2 = \frac{(v^T W u)^2}{\sqrt{(u^T W^T W u)(v^T W W^T v)}}$ would be a measure of the cumulative association between the selected sets of variants and genes by aggregating multiple (possibly weaker) associations (Fig. 1A shows an illustration using P variants and G genes). Multiple such components (u v), can be extracted to reflect approximately orthogonal latent factors of the aggregative correlation, corresponding to possibly distinct mechanisms of trans-regulation (See Supplementary Note 1).

At suitable levels of sparsity (See Supplementary Note 1), ARCHIE components produce a much smaller number of selected target genes which harbor multiple moderate to weak *trans*-association from a selected set of trait-associated variants, thus reflecting a trait-specific pattern of *trans*-association. A detailed algorithm for the estimation of the ARCHIE components is provided in Supplementary Note 1. We can interpret the ARCHIE output as the subset of genes (selected by the gene component) having possibly multiple weaker trans-associations to the subset of variants (selected by the variant component).

**Testing hypothesis of enrichment of trait-specific *trans*-association using a competitive null hypothesis framework.** To test which ARCHIE components significantly capture the phenotype-specific *trans*-association pattern we evaluated the results from the original analysis against a *competitive* null hypothesis. Since trait-related variants are expected to be enriched for *trans*-eQTLs in general, we test whether the cc-values obtained in the original analysis are higher than that obtained using the *trans*-summary statistics between a random set of GWAS-identified variants and genes of similar size, that do not reflect any trait-specific pattern. For this, we first construct a *null matrix* by taking a random sample of $p$ variants from the pool of all variants available and extracting the corresponding *trans*-summary statistics for another set of randomly chosen g genes. Since eQTLGen reports the *trans*-summary statistics across about 10,000 variants associated with different traits, we can construct the *null matrix* using the *trans*-summary statistics from these variants that are associated with different traits and not with the trait of interest. This matrix of *trans*-associations, by design, should not reflect phenotype-specific patterns. For example, in the analysis for SCZ using summary statistics across 218 variants and 7047 genes, we construct the null matrix using 1 variant selected at random from 218 randomly chosen traits and extracting their corresponding *trans*-summary statistics across 7047 randomly chosen genes.

Then we use ARCHIE with the same sparsity levels as the original analysis, to extract the gene and variant components and calculate corresponding cc values. We repeat this step multiple (M) times to generate a competitive null distribution of cc values. We then evaluate the observed cc-values from the original analysis against the corresponding competitive null distributions to calculate the p-value. In particular, the $p$ value of the $k$th ARCHIE component is given as:

$$p_k = \frac{\sum_{i=1}^{M} I(q_k^2 > q_{k;null(i)}^2)}{M} \quad (5)$$

where $q_k^2$ denotes the $k$th cc-value in the original analysis and $q_{k;null(.)}^2$ denotes the elements of the null distribution of the $k$th cc-value. We declare that the top L components significantly capture phenotype-specific trans-association patterns if

$$L = \min\{k : p_k > \alpha; k = 1, 2, \ldots, \min(p, g)\} - 1 \quad (6)$$

The random set of $p$ variants should be carefully chosen so that none of the variants associated with the phenotype in consideration or any phenotype sharing substantial genetic correlation, are included. Further, the set should be such that it does not include a large fraction of the variants from the same phenotype (different from the original phenotype), which may bias the *competitive* null distribution towards the *trans*-association cc-values for that phenotype.

**Simulation model.** To demonstrate that ARCHIE can identify downstream trans-associations, we simulate individual-level gene-expression data for $N$ individuals ($N = 1000$ or 30,000). First, we randomly sampled N unrelated individuals with genotypes at 50 independent and randomly selected SNPs from the UK Biobank with minor allele frequencies ranging from 10 to 40%. The SNPs were then arranged in sets of five each. Each set of five SNPs was then used to simulate the cis-gene expression of eight genes in total marked in red (Fig. 2A–C). For type-I error simulations (Fig. 2A) the expressions for genes 1–4 were simulated from a univariate standard Gaussian distribution and the subsequent downstream genes 5–9 were simulated using the causal regulatory model. The gene-gene regulatory causal effects on genes 5–9 were chosen such that the ~20–30% of the variation in gene expression was explained by causal upstream genes.

For power simulations (Fig. 2B, C), the direct cis-regulatory effects of the five SNPs on the corresponding cis-gene expression were chosen from a Gaussian distribution such that the average cis-heritability explained by the SNPs were maintained at ~20–22%. Then we simulated gene expression for genes 1 through 9 using a causal regulatory model as shown in Fig. 2B, C, using the simulated cis-gene expressions. The regulatory effects between the gene expressions were chosen such that the total variance explained for the expression of Genes 1–9 by indirect effect of the SNPs (trans-heritability) were ~10–14%. For the global null scenario, the downstream genes 1–9 were simulated independent from the cis genes (Fig. 2A) using the causal regulatory model. We further simulated a separate sample of 700 ($N_{Ref}$) from which the gene expressions were used to estimate the coexpression

matrix ($\Sigma_{EE}$). The $\Sigma_{GG}$ matrix was taken to be a diagonal matrix with the diagonal elements being the estimated sample variance of the SNPs. Using this simulated data, we applied ARCHIE and compared the results to that obtained from standard trans-eQTL mapping. To test against the global null hypothesis, that there was no trans-association between the SNPs and any gene in the gene network, we simulated null $\Sigma_{GE;null}$ as $\mathbf{vec}(\Sigma_{GE;null}) \sim N(0, \Sigma_{EE} \Sigma_{GG})$ where vec(.) represents the stacked row vectorization of a matrix and  represents the Kronecker product. We simulated applied ARCHIE to the $\Sigma_{GE;null}$ multiple times to obtain a distribution of the sparse canonical correlation scores ($q^2$). Any ARCHIE component with p-value less than the chosen level was declared significant and the corresponding selected genes were noted (See Supplementary Note 2 for more details).

Next, through resampling experiments we assess whether ARCHIE can potentially identify trait specific trans-associations. Using data from eQTLGen consortium, we construct a matrix of trans-eQTL summary statistics (Z values) across for p variants and g genes. Out of the p variants, we set δ proportion of them to be related to a particular trait. For high values of δ we expect that the trans-summary statistics matrix would reflect trans-association patterns pertaining to the trait and hence should be captured by the ARCHIE components. We then applied ARCHIE on this matrix and evaluate the significance of the components. We replicate this experiment multiple times in a given setting, to estimate the empirical probability of at least one ARCHIE component to be significant (See Supplementary Note 2 for details) and reported the empirical probability of at least one ARCHIE component to be significant across varying values of δ. We repeated this resampling experiment with four different traits (Supplementary Fig. 4).

**Analysis of eQTLGen data**. To identify phenotype-specific trans-associations, we applied ARCHIE on the trans-association summary statistics for 10,317 trait-related variants across 19,942 genes reported by the eQTLGen consortium[19] (See Sample Description for details on the study). In line with the consortium, we defined any gene to be trans to a variant if the variant was located at least 5 Mb from the transcription start site of the gene or on another chromosome. The data contains multiple variants associated with the same trait analyzed for trans-eQTL mapping. Our analysis was restricted to phenotypes that had at least 100 associated variants tested for trans-mapping in the consortium, producing 29 phenotypes. Figure 1B shows a graphical representation of the major steps of our workflow. Briefly, for each phenotype, we extracted the summary trans-eQTL association statistics (Z-score, p-value) and removed all genes that were in within 5 Mb of any of the trait-related variants. In the preprocessing step, we filtered for any missing data and retained the genes that were also expressed in GTEx (v8)[28] whole blood. This produced a list of approximately 129 (min: 112; max: 533) variants and 10,219 (min: 3426; max: 13910) genes on an average per phenotype. ARCHIE requires two additional matrices representing the correlation among the variants themselves (a linkage-disequilibrium matrix) and among the gene-expression levels (a co-expression matrix), which can be estimated using reference data. We constructed the LD-matrix for the variants from individual-level genotype information using 5,000 randomly selected, unrelated European samples in UK Biobank[80]. For the correlation between gene-expressions, we used a penalized co-expression matrix[81] of the corresponding genes constructed from the covariate-adjusted quantile normalized gene-expression levels for individuals in GTEx v8 data. Subsequently, for the given trait, we extracted the selected variants and genes using the significant components and were evaluated for presence of false-positives due to cross-mapping.

**Cross-mappability**. Alignment errors due to similarity in sequenced reads can lead to a substantial rise in false positives for detecting trans-eQTL associations[82]. With the selected ARCHIE components, we extracted the nearby genes expressed in GTEx v8 whole blood for the selected variants (TSS within ±500 kb of the variant) and evaluated the cross-mapping scores for these genes with the selected target genes. Across the 3 traits analyzed in this article, we found that all such gene pairs were mostly non cross-mappable (SCZ: 99.98%, UC: 99.17%, PC: 99.93%), indicating that the trans-association patterns were less likely to be affected by false positive arising from alignment errors.

**Quantifying and testing for enrichment for trait heritability explained by identified target genes**. In the following, we propose a method for quantifying trait heritability explained by the GWAS variants that would be mediated by the identified target-genes and develop a corresponding test for enrichment through comparison of such estimates of mediated heritability associated with that from random genes. For or a particular trait of interest, we start with the Z scores for regression-based trans-eQTL mapping for a set of underlying p variants and g genes. We will assume that, using ARCHIE, we have identified G target genes that capture trait-specific trans-association patterns. To perform the test as proposed above, we require individual-level phenotype and genotype data independent of the samples used in the original analysis. Given genotypes (or dosages) at the p variant sites for an individual k, for each target gene, we define the trans-imputed expression scores (TIES) as the predicted expression value for the jth target gene as

$$\text{TIES}(p)_{jk} = \sum_{i=1}^{p} \frac{Z_{ij} x_{ik}}{\sqrt{2m_i(1 - m_i)}}, \quad (7)$$

where $Z_{ij}$ is the z-score for the effect of the ith trait-related variant on the jth gene, $x_{ik}$ is the genotype or dosage for the kth individual at the ith variant and $m_i$ is the minor allele frequency of the ith variant. We construct the TIES under two different schemes:

1. Using all the trait-related variants with complete trans-association statistics reported in eQTLGen.
2. Using only the trait-related variants selected in the significant components.

To evaluate how strongly the TIES for the G target genes are associated with the phenotype levels, we use the following multiple regression model

$$g[E(y_k)] = \beta_0 + \sum_{j=1}^{G} \beta_j \text{TIES}(p)_{jk} \quad (8)$$

where $y_k$ is the phenotype value (e.g., disease status) for the kth individual; g[.] is a canonical link function and can be set to be the identity function for continuous phenotype or the logistic function for binary (disease status) phenotypes. We record the pseudo-$r^2$ from this regression model as a measure of association between the TIES and the phenotype value. The pseudo-$r^2$ would provide an estimate of trans-heritability, meaning it can quantify the variance explained by the trait-related variants that is expected to be mediated via the selected target genes in the context of the trans-associations reported. To test whether the observed $r^2$ is significant in comparison to what is expected at random, we adopt a resampling-based approach. We sampled g genes (excluding the originally selected target genes) from the genome, constructed the corresponding TIES for the individuals and recorded the $r^2$ for the regression model. This would a null estimate of trans-heritability of the SNPs expected to be mediated by a set of g genes. If the observed pseudo-$r^2$ is substantially higher than the null estimates, we can infer that the trans-associations selected by ARCHIE explain higher variance compared to that expected through random trans-associations. We performed resampling multiple (1000) times to generate a control (null) distribution of $r^2$ to reflect the associations expected from random genes. We then calculated the p value of the observed $r^2$ using the originally selected g genes from this control distribution to evaluate whether the TIES have any significant association with the phenotype.

Approximately, the observed $r^2$ reflects the proportion of trait-variance explained by the TIES. Thus, a significantly higher $r^2$ would imply that the selected genes harbor several trans-associations and mediate the effects of the trait-related variants more than any random set of genes. As the analysis of association between TIES and trait (for both the selected trans genes and random genes) is performed in an independent dataset, and no information on directions or magnitudes of trait association for the SNPs are used in the original ARCHIE analysis, the test for heritability enrichment provides independent validation of the relevance of selected target-genes in explaining genetic associations for the trait. In our application, we used individual-level phenotype and genotype data from UK Biobank participants to estimate association between TIES and traits.

We also performed several other follow-up analyses including PPI enrichment, pathway enrichment, and differentially expressed genes enrichment. These analyses were carried out using pre-established standard pipelines. For full details on these see Supplementary Note 3.

**Reporting summary**. Further information on research design is available in the Nature Research Reporting Summary linked to this article.

## Data availability

The eQTLGen consortium summary statistics for trans-eQTL associations were obtained from: https://www.eqtlgen.org/trans-eqtls.html (downloaded on 09/01/2019). The UK BioBank data were obtained from https://www.ukbiobank.ac.uk under the UK BioBank resource application 17712. GTEx data were obtained from dbGAP (accession id: phs000424.v8.p2). For details on accessing and interpreting GTEx data please see: https://www.gtexportal.org/home/. Individual-level data for the DGN cohort are available by application through the NIMH Center for Collaborative Genomic Studies on Mental Disorders. Instructions for requesting access to data can be found at https://www.nimhgenetics.org/access_data_biomaterial.php, and inquiries should reference the "Depression Genes and Networks study (D. Levinson, PI)". Individual-level genotype data from 1000 Genomes study can be accessed from: https://www.internationalgenome.org/data/. The summary data, results generated through this study and source data for figures are provided in the GitHub repository: https://github.com/diptavo/ARCHIE and in the Zenodo database[83] at: https://doi.org/10.5281/zenodo.6533206. Source data are provided in this paper.

## Code availability

The codes and an example simulated data can be found on ARCHIE GitHub repository: https://github.com/diptavo/ARCHIE and in Zenodo[83] database: https://doi.org/10.5281/zenodo.6533206.

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

## Acknowledgements

N.C., D.D, and A.B. were supported by NIH R01-HG010480-01. A.B. was additionally supported by 1R01MH109905 (NIMH).

## Author contributions

D.D., N.C., and A.B. conceived the project. D.D. obtained the data and carried out the data analysis. D.D., Y.H., A.S., M.A., N.C., and A.B. interpreted the results. D.D. wrote the manuscript under the supervision of N.C. and A.B. All the authors critically read and approved the manuscript.

## Competing interests

A.B. is a consultant for Third Rock Ventures, LLC and a shareholder in Alphabet, Inc. The other authors declare no competing interests.
