## [Peer Review File · Nature Communications]

Aggregative trans-eQTL analysis detects trait-specific target gene sets in whole bloodREVIEWER COMMENTS

Reviewer #1 (Remarks to the Author):

This paper presents a statistical framework to identify trans genes relevant to a complex trait based on trans-eQTL mapping summary statistics. The application of sparse CCA in this context is interesting. Even though the real data analysis results are interesting, the simulation study is far from adequate. I have the following concerns.

1. The simulation study in the paper is very limited, and leaves substantial room for speculation of the overall merit of the method. For example, what is the specificity and sensitivity of the selected subset of target genes? Studying the accuracy of the selected subset of genes is crucial in this paper. Otherwise, picking too many false target genes may not help much over trans-eQTL mapping. The simulation study was done only for few genes, and only one possible gene network. So, for careful evaluation of the method, more extensive simulation study needs to be performed.
2. There can be many other possible networks of the genes in the simulation study. I wonder if good results are robust to the choice of such a network.
3. What about type 1 error rate while testing $\delta = 0$? No results were reported.
4. What is the impact of the p-value cut-off in the trans-eQTL analysis? In particular, it becomes very difficult to compare the two approaches with respect to power, since there is this concern of the choice of the p-value threshold. Also, if we really want to say, one is more powerful, the two approaches should be compared with respect to the same choice of the rate of type 1 error.
5. How are the variants associated with the phenotype are selected? Is it all SNPs that cross the GW cut-off of association, or the ones after LD-pruned? Also, as discussed in the paper, the method does not involve the variant trait association statistics. The authors should discuss at least one possible way of doing that.
6. Is there any other reference for sCCA based on summary statistics? Discuss how the estimation technique is different from the one proposed by Witten et al.
7. Will there be any issue with the sCCA estimation if the SNP LD matrix is obtained from the 1000G data? Similar penalized estimate of covariate matrix as used for gene-expression matrix could also be used for LD matrix. Otherwise, we need individual-level data anyway, at least one cohort with large sample size, and the method loses the advantage of only requiring summary statistics.
8. In the differential enrichment analysis, when making the list of background differential genes for a specific tissue, p-value cut-off used was 0.05 without any correction for multiple testing?
9. In real data analysis, both for UC and SCZ, identified target genes are enriched among immune related pathways. SCZ being far from an immune-related disease, it sounds odd.

10. The claim that the trans-eQTL mapping is less powerful may not be appropriate. Because, the two approaches control different type of error rates.
11. It seems that the set of genes analyzed are trans to the set of the GWAS significant variants. Otherwise, a variant trans for a gene can be cis for another gene. This needs to be clarified. Some clarification is needed.
12. How is $\Sigma_{\{ge\}}$ obtained based on summary statistics? More details need to be provided.
13. In general, Elastic net penalty can be more effective than Lasso penalty with respect to selection accuracy, in particular, in presence of arbitrary correlation structure. Is it difficult to be implemented based on summary statistics? Some discussion will be helpful.
14. Since, orthogonal axes are not identified by the estimation procedure used in sCCA, does it increase the chance of selecting variants that are in LD, or trans genes that are co-regulated?
15. sCCA will select some suggestive SNPs and genes to have trans associations. However, how different are those obtained from trans eQTL mapping simply by using relaxed p-value threshold or using FDR approach?
16. Some choices in the simulation design are strange: MAF 20%-35%, why not 5%-45%? $\beta_i \sim N(0.7, 0.1)$; $\gamma_i \sim N(0.5, 0.1)$. Do all positive choices of the means favor the method somehow?
17. Any justification why some specific tissues are enriched in the differential expression analysis for a given phenotype, but not the other tissues? Is there any evidence that the tissues which are possibly more relevant for the phenotype are also being found to be enriched?
18. Why the associations for gene 5 and 6 arise due to pleiotropy in the figure presenting gene network in the simulation study? More explanation is needed.
19. In the main methods section of the paper, motivation of the method starting with individual-level data need to be included.
20. There are many typos in the paper.

Reviewer #2 (Remarks to the Author):

Review of Dutta et al. - Aggregative trans-eQTL analysis detects trait-specific target gene sets in whole blood

In this study, Dutta and colleagues develop a novel method to identify trait-specific trans-eQTLs and, as a result, prioritize trait-relevant genes. The authors use several publicly available datasets to show that it is possible to identify genes that were not found before using regular trans-eQTL mapping and they illustrate the relevance of the novel prioritized genes with several enrichment and follow-up analyses.

The methodology of this paper is clever and it is a useful contribution on top of existing methods. However, the manuscript and in particular the figures could use editing for clarity and to emphasize the main message of the article. I also have some questions regarding the novelty of the results and some methodological details.

Major comments

- Fig 1A could use more information and description within the figure panels, e.g. it is not immediately clear what is contained in the heatmaps (trans-eQTL summary Z-scores? P-values?). The flow of the diagram would be better if it went from left to right and/or top to bottom with a panel in between the 2 heatmaps that outlines the things ARCHIE does ('selects a subset of variants and genes') and data it uses (the LD matrix and co-expression from external datasets). You could even use Supp Fig 1 for this.
- In the section 'Identification of downstream genes' please start by explaining that you are simulating a network with 9 genes and which of those you considered causal (all 9? Only 5-9?). Are genes 1-4 supposed to be local genes? Please also describe the difference of genes 5 and 6 as opposed to genes 7-9; I suppose the point is that they are all downstream effects but only 7-9 are affected by convergence of multiple significant genes, but that is not clear from the text. Also, 9 genes are a pretty low number for a set of causal genes, it would be good to simulate with more genes.
- Were the novel genes identified by ARCHIE, e.g. the 59 for schizophrenia, tested in eQTLGen? And if so, do the ARCHIE genes and the non-significant eQTLGen trans-eQTLs have the same direction of effect?
- How many of the novel ARCHIE genes are identified as being correlated to the polygenic risk score for the same trait? The eQTLGen consortium provides such correlation estimates ('eQTS') which sound like they should give similar results to your ARCHIE genes.
- Were the genes that you specifically highlight in the text found in eQTLGen already or not? (CXCR4, DISC1, GPA33, NFKB1, etc.)
- Where were the SNPs located on the genome? E.g. were the 9 variants for prostate cancer spread across multiple chromosomes or are they located in one or two loci?
- You mention in the discussion that the direction of the effect of the SNPs on the trait is not incorporated into ARCHIE. Why not? It seems that the results would become much more interpretable if you use this information, because right now you aggregate trans-eQTLs from both risk and protected variants into your model. For simplicity, could you just subset the risk variants and run ARCHIE on those SNPs?
- Is there a way to include cis-eQTLs in ARCHIE? In figure 1B, the assumption is that gene expression levels of genes 5-9 are (partially) mediated by the gene expression levels of cis-genes 1-4. Could that increase the power to detect causal genes? Especially since some of the genes in the examples are located within GWAS loci.
- What is the effect of using a co-expression matrix from a reference panel (GTEx)? Why did you not use the co-expression from the eQTLGen consortium directly?

- “we found the selected trans-associations, in particular, the target genes mediate significant trait heritability (p -value < 0.001) than expected by chance” ☒ this sentence needs a re-write to become understandable. The enrichment of trait heritability is a nice method that, if I am not mistaken, is novel. I think you can emphasize this method and its results more in the main text. Also, figures 2-4D would be stronger if you add the estimated trait heritability using significant trans-eQTL SNPs for both the trait under investigation (e.g. SCZ, UC, prostate cancer) as well as another trait (or the results of the resampling method), which would be lower than the ARCHIE selected SNPs. Right now, the message of this picture seems to be: ‘ARCHIE selected SNPs contribute less to heritability than all SNPs associated to this trait’, while I suppose that you want it to say: ‘ARCHIE selected SNPs are specifically enriched for heritability of the trait we are investigating’.

Minor comments

- Although I understand the desire for a catchy name, ‘ARCHIE’ is far from an acronym. At this point, I suggest to either just call it ‘ARCHIE’ without highlighting random letters in the words or to come up with an actual acronym.
- There are quite a number of mistakes in verb conjugation, e.g. “ARCHIE selects the downstream genes (Genes 7-9) that has trans-associations” should be “ARCHIE selects the downstream genes (Genes 7-9) that have trans-associations”. Please have a careful look at the grammar throughout.
- “The selected genes are broadly trans-regulated by the selected SNPs and mediate their effect to the trait.” ☒ This statement is unsubstantiated. You cannot be sure that the genes mediate the trait without testing for causality or mediation analyses.
- “we found that ARCHIE selects the downstream genes (Genes 7-9) that has trans-associations to multiple upstream genetic variants, with high probability” ☒ please add the actual probability here
- What was the rationale for using sample sizes of 1,000 and 30,000 in the simulation studies?
- How did you select the 3 traits with more in-depth results?
- Please briefly describe the ‘competitive null distributions’ in the legends for figures 2, 3 and 4.
- Please indicate in figures 2-4A of the ARCHIE components which ones are significant and which are not.
- Please add a legend for the colors in Fig. 2C and 3B. It is also a bit confusing to plot 2 variables on the x-axis, it would be clearer if you plot only the p-value and color the bars by % overlap or vice versa.
- Please label the y-axis of Fig. 3C.
- Please label the axes of Supp Fig 2

Reviewer #3 (Remarks to the Author):

Review:

Aggregative trans-eQTL analysis detects trait-specific target gene sets in whole blood

Overview:

Diptavo et al. introduce a new method called Aggregative tRans assoCiation to detect pHenotype specific gEne-sets (ARCHIE) to detect the trans associations between a set of known trait-related genetic variants and a set of co-regulated genes. This method is motivated by the omnigenic model which predicts that the majority of SNP-heritability is mediated by pervasive trans-regulatory effects, which are historically challenging to identify due to limited statistical power. The core inferential framework of ARCHIE is sparse Canonical Correlation Analysis coupled with a permutation test to identify trans-regulatory networks limited to a set of known trait-associated genetic variants. A key novelty is the ability of ARCHIE to run directly on GWAS and eQTL summary statistics, provided estimates of population LD and gene co-expression from publicly available datasets (1000G, GTEx, etc). Applying the ARCHIE model GWAS summary statistics from 29 traits together with eQTL summary data from eQTLGen, they identify many novel genes which are supported with gene set enrichment/ontology analyses.

Overall, I found the paper to be well written, highly relevant, and the statistical methodology sound. With that said, I have a few comments related to simulations and real-data replication.

Major Comments:

1. I appreciate the use of simulations to demonstrate the feasibility of ARCHIE under ideal circumstances and thank the authors for their inclusion, however I have three primary concerns that could be addressed with expanded or altered simulations.

a. First, I find the parameters used to be a bit arbitrary and likely unreflective of actual biology. Can the authors re-parameterize cis- and trans-effect sizes to be proportional to cis-h2g or trans-h2g estimates? For example if the cis-h2g of a given upstream/core-gene is 0.1 and there are 5 SNPs with direct effects, then (assuming no LD) $\beta \sim N(0, 0.1 / 5)$. If trans-regulatory effects are due to distal cis-effects, then the mediating effect can be 'reverse-engineered' from the cis-h2g at the cis-gene(s). This would help place findings and statistical power into a realistic context.

b. Second, the only simulation performed was under the alternative where trait-specific trans-regulatory action is occurring on genes measured in the relevant tissue. Adding simulations under the null demonstrating that p-values are well-calibrated would greatly help interpretation. There is some

numerical evidence presented to this using real-data results, but a clearer link to calibration and QQ plots would be helpful.

c. Last, it would be helpful to see how ARCHIE performs in simulations with model misspecification regarding the use of external Sigma_EE matrices. For example, the regulatory/generative model could be fixed to generate Sigma_GG, Sigma_GE matrices, but a noisy (sampling around some true Sigma_EE using Wishart) or incorrect Sigma_EE (relevant regulatory networks are only partially captured/correlated due to “non-relevant” tissue) is used in the inference procedure.

2. I’m glad the authors performed several enrichment analyses to provide support for their gene lists, however no thorough perturbation check was performed on using publicly available data as a means to estimate the correlation matrices. Are the real-data results robust to replacing Sigma_EE estimated in GTEx whole blood expression with say, whole blood expression from DGN (at least within genes overlapping both studies) or some reasonable alternative?

Minor Comments:

1. What is the alpha threshold used to determine power for simulations/Figure 1C?

2. Why was a significance threshold of 0.001 used in Supplementary Figure 2?

3. The regression formula at the bottom of page 27 seems to have a typo. Should there be ‘beta_j’ for each of the G TIES(p) values?

Reviewer #1 (Remarks to the Author):

This paper presents a statistical framework to identify trans genes relevant to a complex trait based on trans-eQTL mapping summary statistics. The application of sparse CCA in this context is interesting. Even though the real data analysis results are interesting, the simulation study is far from adequate. I have the following concerns.

We thank the reviewer for this comment. We have now further broadened the scope of our simulation studies. Specifically, we have studied the type-I error of ARCHIE in comparison to standard trans-eQTL mapping and compared the power of the two methods under several different causal network structures. Further, we have also assessed the sensitivity and specificity of the genes selected by ARCHIE in comparison to standard trans-eQTL mapping.

1. The simulation study in the paper is very limited and leaves substantial room for speculation of the overall merit of the method. For example, what is the specificity and sensitivity of the selected subset of target genes? Studying the accuracy of the selected subset of genes is crucial in this paper. Otherwise, picking too many false target genes may not help much over trans-eQTL mapping. The simulation study was done only for few genes, and only one possible gene network. So, for careful evaluation of the method, more extensive simulation study needs to be performed.

As mentioned above we have now expanded the simulation study in the manuscript to compare the performance of ARCHIE in terms of type-I error and power.

Type-I error: ARCHIE addresses a composite null hypothesis that there is no association between a group of variants and genes, while standard trans-eQTL mapping has the simpler null hypothesis that there is no association for any individual variant-gene pair. Hence the concept of type-I error rate would be quite different for each of them. However, for a fair comparison, we have also demonstrated the type-I error of ARCHIE under the global null hypothesis that there exists no association between any variant-gene pair. The figure below shows a simulation scenario under the global null hypothesis where a set of 40 SNPs do not have any association with a network of 9 genes.

Hypothesis	Method	Level			
		1×10^{-3}	1×10^{-4}	1×10^{-5}	1×10^{-6}
At least one variant-gene association significant	Standard trans-eQTL	0.15	0.044	0.004	2.1×10^{-4}
At least one ARCHIE component significant	ARCHIE	1.8×10^{-4}	9.1×10^{-6}	3.9×10^{-7}	2.5×10^{-7}

Under this global null scenario, the SNPs 1-40 do not have any association with the network of 9 genes. The type-I error rate for ARCHIE is defined as the proportion of simulation iterations where the p-value for at least one ARCHIE component is less than the level. Correspondingly for standard trans-eQTL analysis, the type-I error is calculated as the proportion of simulation iterations where there is at least one p-value is less than the level for all possible variant-gene pairs.

As demonstrated in the above table, ARCHIE maintains a conservative type-I error even at nominal levels such as 1×10^{-3} or 1×10^{-4} . In comparison, the overall familywise error rate of standard trans-eQTL mapping is maintained at 0.05 at the approximate Bonferroni correction level for 40 SNPs and 9 genes ($0.05/(40 \times 9) = 1 \times 10^{-4}$ approximately). At a further relaxed threshold (1×10^{-3}), standard trans-eQTL has higher false positives. The type-I error rate of ARCHIE for the same level is much lower.

Thus, to compare the power of ARCHIE and standard trans-eQTL mapping, we need to achieve similar type-I error rates. This can be done by making the level for ARCHIE lenient. We find that approximately similar type-I error rates are achieved at a level of 9×10^{-4} for ARCHIE and 1×10^{-6} for standard trans-eQTL mapping. Thus, we use these levels in further analysis of power (see below).

The above discussion has been included in the main manuscript in Section Results (subsection Simulation Study Results: Comparison with Standard trans-eQTL Analysis), Lines 124-138. The Figure demonstrating the global null scenario is incorporated as Figure 2A. The results for the type-I error simulation are now in Table 1. Detailed information about the simulation models is outlined in Methods (Lines 911-933) and Supplementary Section B: Numerical Experiments.

Power: By calibrating for similar type-I error rates respectively (See above discussion), we

now estimate the power of ARCHIE in comparison to standard trans-eQTL mapping under the two causal networks shown in Figure. The details of the simulation settings and the choice of the parameters have been outlined in Section: Simulation Model and Supplementary Methods. Briefly, the SNPs 1-40 are independent with MAF between 10-40%; the cis gene expressions (marked in red) have 20-22% of the cis heritability explained by the 5 corresponding SNPs; the distal gene expressions (marked in blue) have 10-14% of the trans heritability explained by the SNPs.

Under these settings, we found that, for the sparse causal network scenario (A), the power of ARCHIE is comparable or slightly more than that of standard trans-eQTL. This is because, in a sparse network, the gene-gene regulatory effects are stronger at a given level of trans-heritability. Stronger regulatory associations are captured by ARCHIE and standard trans-eQTL mapping with similar probability.

For dense causal network scenario (B), the power of ARCHIE is substantially more than that of standard trans-eQTL. This is because, in the denser regulatory network, gene-gene effects are weaker and standard trans-eQTL is underpowered to identify such effects. Conversely,

ARCHIE can effectively aggregate multiple weak associations converging on a gene and select the pertinent gene network.

The above discussion has been included in the main manuscript in Section Results (subsection Simulation Study Results), lines 140-191. The Figures demonstrating causal regulatory models and the estimated power are incorporated as Figure 2B-E. Detailed information about the simulation models and modeling parameters is mentioned in Methods (Lines 911-933) and Supplementary Section B.

Further, we have assessed the power of ARCHIE in comparison to standard trans-eQTL in presence of a master regulator gene (Supplementary Section B2 and Supplementary Figure 2). The overall results remain similar, in that ARCHIE maintains a comparable or higher power compared to standard trans-eQTL mapping in detecting downstream target genes.

Thus, in summary, we find that ARCHIE has higher power than standard trans-eQTL analysis when both methods are calibrated to maintain the same type-I error under the global null hypothesis of no association.

However, we note that a primary objective of our analysis was to detect a disease (or trait)-specific trans-association pattern in the background of broader trans-associations that are expected to be seen in the genome. Thus, we considered the “*competitive null hypothesis*” to be more pertinent for our main analysis and we have demonstrated type-I error and power of ARCHIE under this type of hypothesis (see Section Results: Assessing trait-specificity, Lines 198-257). Standard trans-eQTL mapping is not suitable for testing under the competitive null hypothesis.

We thank the reviewer for this comment on the simulations which have motivated us to extensively increase the scope and the depth of the simulation studies. The methods and results are described in detail in the manuscript in the Sections: Results, Simulation Model, and Supplementary Section B.

2. There can be many other possible networks of the genes in the simulation study. I wonder if good results are robust to the choice of such a network.

This is indeed true that the space of alternative hypothesis is extremely large. We have now included several different types of alternate models: sparse gene network (Figure 2B, D), dense gene network (Figure 2C, E), and gene network with master regulator (Supplementary Figure 2). For each of the scenarios, we find that ARCHIE is powerful for detecting gene networks that are regulated in trans by genetic variants.

We acknowledge that there could be many other possibilities of the intermediate causal mechanism and it is beyond the scope of the current article to comprehensively evaluate all (or a majority of) models. However, the principal goal of the simulations was to indicate the nature of the causal structure that can potentially be identified by ARCHIE and in which possible scenarios it can outperform standard trans-eQTL mapping. We have acknowledged this in the text:

Section Results. Line 193-196.

“It is to be noted that the potential space of causal gene networks extremely large, and a comprehensive evaluation is beyond the scope of this article. However, the above small-scale simulations provide us with an intuitive insight on which causal structures might be detected by ARCHIE.”

3. What about type 1 error rate while testing $\delta = 0$? No results were reported.

Thank you for the chance to clarify this point. This is pertaining to the type-I error rate corresponding to the competitive null hypothesis, which tests whether the SNPs and genes identified by ARCHIE components have a higher sparse canonical correlation than what is expected from random GWAS SNPs, i.e., SNPs associated with other diseases, identified by GWAS studies. This represents a stringent null hypothesis as we know that in general GWAS SNPs are enriched in trans-eQTLs.

For $\delta = 0$, meaning the GWAS SNPs included in ARCHIE analysis are completely non-trait specific and chosen at random, the proportion of resampling iterations where ARCHIE components are declared significant would correspond to the type-I error of ARCHIE for the competitive null hypothesis. We have now increased the resampling iterations to 100,000 to accurately estimate the competitive type-I error of ARCHIE. ARCHIE maintains a conservative type-I error rate for this across different thresholds.

We have now included this result in the text (Lines 252-257) as well as the Supplementary Table 1.

Section Results. Lines 252-257

“In particular, at $\delta = 0$, the results would correspond to the type-I error of ARCHIE under the competitive null hypothesis. We found that at a level of 1×10^{-04} , the competitive type-I error of ARCHIE is conservative for all the traits under consideration indicating that ARCHIE produces reduced false positives (Supplementary Table 1).”

4. What is the impact of the p-value cut-off in the trans-eQTL analysis? In particular, it becomes very difficult to compare the two approaches with respect to power, since there is this concern of the choice of the p-value threshold. Also, if we really want to say, one is more powerful, the two approaches should be compared with respect to the same choice of the rate of type 1 error.

See our responses to comment 1 above.

5. How are the variants associated with the phenotype are selected? Is it all SNPs that cross the GW cut-off of association, or the ones after LD-pruned? Also, as discussed in the paper, the method does not involve the variant trait association statistics. The authors should discuss at least one possible way of doing that.

Thank you for raising this point. The eQTLGen consortium analyzed a list of SNPs that have been reported to be associated with different traits or diseases in previous large GWAS, curated from various sources including the GWAS catalog, Immunobase, and others. The eQTLGen authors provide trans-eQTL summary data on approximately 10,000 such variants. Although there are some instances where there are more than one SNP analyzed in one locus and has a strong LD between them, in general, the SNPs represent independent loci.

We have clarified this further in our manuscript (Lines 249-251).

“The eQTLGen consortium provides summary statistics (Z-values, p-values) from standard trans-eQTL mapping for more than 10,000 genetic variants across numerous loci on the genome, curated from external databases, that have been identified to be associated to traits and diseases in large scale GWAS.”

Further, we have now included a short discussion about incorporating variant-trait association statistics as follows:

Section Discussion. Line 527-532.

“However, it is likely that incorporation of the GWAS effect sizes (value and direction) of trait association for the SNPs in the sCCA itself will lead to improved power for detection of the trait-specific target genes. One of the immediate ways to incorporate that might be to weight the variants with weights proportional to squared (or absolute value) of the GWAS effect size. However, incorporating the direction of the GWAS effect with the sCCA framework remains an interesting problem and merits further research.”

Also, see the response to Reviewer 2 Comment 7 and 8.

6. Is there any other reference for sCCA based on summary statistics? Discuss how the estimation technique is different from the one proposed by Witten et al.

We have now included the following references on the use of sCCA from genomic, transcriptomic, and neuroimaging studies:

Rosa, M. J. *et al.* Estimating multivariate similarity between neuroimaging datasets with sparse canonical correlation analysis: An application to perfusion imaging. *Front. Neurosci.* (2015).

Feng, H. *et al.* Leveraging expression from multiple tissues using sparse canonical correlation analysis and aggregate tests improves the power of transcriptome-wide association studies. *PLOS Genet.* 17, e1008973 (2021).

Rodosthenous, T., Shahrezaei, V. & Evangelou, M. Integrating multi-OMICS data through sparse canonical correlation analysis for the prediction of complex traits: a comparison study. *Bioinformatics* 36, 4616–4625 (2020).

As a part of the methods and applications of matrix decomposition, Witten et al (2009) had introduced an elaborate work on sparse canonical correlations (sCCA) and their applications on genetic datasets. However, our current method ARCHIE has two important differences with that:

1. ARCHIE corrects for row and column covariance matrices and identifies sets of variants that have possibly multiple independent trans-associations with sets of genes while Witten et al recommend assuming a diagonal row (or column) covariance structure.

2. The estimation and testing procedure of ARCHIE is formulated in terms of publicly available summary statistics.

Further, the application of Witten et al was to identify sets of genes whose expressions are most correlated with genotypes in samples in the CGH study. The approach was agnostic of the physical location of the genes and hence the mechanisms are not separately interpretable as the cis and trans regulation.

We have included a short discussion on these in Supplementary Section A1.

7. Will there be any issue with the sCCA estimation if the SNP LD matrix is obtained from the 1000G data? Similar penalized estimate of covariate matrix as used for gene-expression matrix could also be used for LD matrix. Otherwise, we need individual-level data anyway, at least one cohort with large sample size, and the method loses the advantage of only requiring summary statistics.

If genotype or dosage data on large cohorts are not available, commonly used reference panels like 1000 Genomes, can be used to estimate the LD matrix (Σ_{GG}) between the SNPs. To estimate the Σ_{GG} matrix, two strategies can be employed:

1. SNPs on separate chromosomes can be considered independent and the LD for SNPs on each chromosome can be estimated separately. In eQTLGen datasets, usually, this strategy results in the number of samples being much larger than the number of SNPs on each chromosome. This guarantees that the Σ_{GG} matrix remains positive semidefinite.

2. If the number of SNPs on each chromosome exceeds the sample size of the reference data, sparse LD estimation with an L1 penalty can be used. This will result in a sparse LD matrix that retains the stronger LD values while reducing the weaker ones to zero. There are several available methods to estimate sparse LD as discussed in Feder et al¹ and Schafer et al².

We have now included this as a discussion in Supplementary Section A3.

8. In the differential enrichment analysis, when making the list of background differential genes for a specific tissue, p-value cut-off used was 0.05 without any correction for multiple testing?

Thank you for pointing this out. To construct the background set of genes for a given tissue, we used an FDR cut off of 0.05 which we call FDR adjusted p-value. We have corrected this omission (Supplementary Section C2).

9. In real data analysis, both for UC and SCZ, identified target genes are enriched among immune related pathways. SCZ being far from an immune-related disease, it sounds odd.

Thank you for raising this question. In general, trans-eQTL associations have been shown to be highly tissue-specific in several analyses including GTEx. Since the eQTLGen reports data on trans-associations in whole blood only, it is expected that ARCHIE will capture trans-regulatory signatures and genes differentially regulated in whole blood. Thus, there would be an excess of immune related or general blood-related pathways that are captured through ARCHIE when using blood data. In the future, ARCHIE could be applied to large-scale data from other tissues including brain tissues, as those data become available. The method will then potentially reveal additional genes and pathways relevant to those tissues.

We note that while for both UC and SCZ enrichment for immune-related pathways is seen, the signals were much stronger for UC. While the connection between immune genes and SCZ is less obvious, we believe this is a noteworthy finding given recent studies have increasingly pointed out complex interactions between the immune system, inflammation, and the brain (Khandaker et al. 2015. The Lancet Psychiatry)³.

We have now added a sentence in the main text (Lines 359-363) pointing out the key review paper on this topic.

10. The claim that the trans-eQTL mapping is less powerful may not be appropriate. Because, the two approaches control different type of error rates.

Please see our responses to comment 1 above.

11. It seems that the set of genes analyzed are trans to the set of the GWAS significant variants. Otherwise, a variant trans for a gene can be cis for another gene. This needs to be clarified. Some clarification is needed.

By design of our analysis, we started with distal genes and SNPs for a given trait, meaning *all* the SNPs in our analysis were more than 5Mb away from the transcription start site of *any* gene. We have now made this clear in the main text.

Section Results. Lines 266-269.

“For each of the 29 traits, we extracted the trans-association summary statistics for the variants associated with the trait and only the genes that were either more 5Mb away from each of the variants or on a different chromosome. Thus, by design, all the genes in the analysis for a trait were distal to all the variants under consideration.”

12. How is $\Sigma_{\{ge\}}$ obtained based on summary statistics? More details need to be provided.

We appreciate this suggestion. Given the summary statistics (Z-value, p-value) of the trans-eQTL mapping, we can estimate Σ_{GE} as:

$$(\Sigma_{GE})_{ij} \approx \frac{Z_{ij}}{\sqrt{2Nm_i(1 - m_i)}}$$

where Z_{ij} is the Z-value for the trans-eQTL mapping (linear regression) of the i^{th} gene expression and j^{th} genetic variant, m_i is the minor allele frequency of the variant and N is the effective sample size. This relationship holds under the assumption that the variance of gene expression explained by the SNP is negligible. In our analysis, Z_{ij} and N are provided by the eQTLGen trans-eQTL summary statistics, and m_i is estimated from an external reference panel like UK Biobank or 1000 Genomes.

We have now added this description in Supplementary Section A3.

13. In general, Elastic net penalty can be more effective than Lasso penalty with respect to selection accuracy, in particular, in presence of arbitrary correlation structure. Is it difficult to be implemented based on summary statistics? Some discussion will be helpful.

We agree with the reviewer that elastic net, or in fact any other regularization penalties can be used in the method. But with increasing penalty parameters estimation algorithm becomes complicated. Further Lasso has high selection consistency⁴. Since we are only interested in selection rather than reducing bias in loadings at this stage, lasso was our choice. However, other regularization approaches, especially elastic net which improves selection under GroupWise or correlated settings, can be useful as well. We have added a discussion on this.

Section Discussion. Lines 537-548.

“Since variable selection is a major goal in ARCHIE, we have introduced regularization via the L1 penalty due to its proven theoretical selection consistency⁴. In fact, in presence of correlation between SNPs as well gene expressions, elastic net regularization approach can be effective in accurate selection. In the future, usefulness of alternative types of penalty functions in the selecting the genes and variants merits further research.”

14. Since, orthogonal axes are not identified by the estimation procedure used in sCCA, does it increase the chance of selecting variants that are in LD, or trans genes that are co-regulated?

Our estimation procedure ensures numerical orthogonality since we choose the sparsity parameters such that the successive axes do not have overlapping selections (for both genes and variants). **See Supplementary Section A2 for further details.** In our analysis, we have been able to estimate the ARCHIE components through this algorithm, but we acknowledge, such solutions might not always be obtainable. In those scenarios, indeed, orthogonality is not guaranteed, although usually approximate orthogonality (correlation between axes is approximately 0) can be ensured through choices of tuning parameters.

We adjust for LD & coexpression, i.e., correlation within the variants and that within the gene expressions as a part of our sCCA algorithm. So ideally sets with multiple independent associations between variants and genes will be selected.

Section Methods. Lines 879-881.

“We can interpret the ARCHIE output as the subset of genes (selected by the gene component) having possibly multiple independent trans-associations with the subset of variants (selected by the variant component).”

15. sCCA will select some suggestive SNPs and genes to have trans associations. However, how different are those obtained from trans eQTL mapping simply by using relaxed p-value threshold or using the FDR approach?

In general, we find the majority (89%) of genes identified through our sCCA-based method (ARCHIE) to have a suggestive nominal p-value (< 0.05) with more than one SNP selected by sCCA. This indicates that sCCA can effectively combine weaker association results.

Section Results. Lines 287-288.

“In fact, a large majority of the novel genes (89%) harbor multiple weaker trans-associations with the variants selected in the variant component.”

Further, from the type-I error simulations, we find that relaxing the p-value threshold for trans-eQTL mapping can result in inflated type-I error meaning higher false discoveries.

This is also evident from the analysis of sensitivity and specificity we performed comparing ARCHIE and standard trans-eQTL. We found that though the sensitivity (selecting non-null genes) is comparable for both, the specificity (not selecting null genes) of ARCHIE is higher compared to standard trans-eQTL, meaning standard trans-eQTL has higher false positives. Thus, relaxing the threshold for standard trans-eQTL might increase the sensitivity but will eventually lower the specificity of the identified genes.

We have included these results and this discussion in the Results Section as well as in the Supplementary Section A5 and Supplementary Figure 3.

Section Results. Lines 188-191.

“We further compared the sensitivity and specificity of ARCHIE with that of standard trans-eQTL and found that although the sensitivity of ARCHIE was only slightly higher than, the specificity was substantially higher compared to trans-eQTL mapping (Supplementary Section A5 and Supplementary Figure 3).”

16. Some choices in the simulation design are strange: MAF 20%-35%, why not 5%-45%? $\text{Beta}_i \sim N(0.7, 0.1)$; $\text{gamma}_i \sim N(0.5, 0.1)$. Do all positive choices of the means favor the method somehow?

We agree in the initial simulations the choices of parameters were somewhat arbitrary. We have now formalized the choices for the parameters (See Reviewer 3 Point 1).

Currently, in all the simulation settings we have included 40 SNPs with MAF ranging from 10% to 45%. A reason for not including lower MAF is that since in one of the scenarios we have only 1000 samples, an SNP with $MAF < 10\%$ might result in a very low minor allele count thus influencing the results and leading to false positives due to outliers or other effects.

For β_i which denotes the association of an SNP with a nearby (cis) gene, we have now chosen the values of β_i from a normal distribution such that the total cis heritability explained by the SNPs for a gene is roughly maintained at 20-22%.

For γ_i which denotes the association between two genes, we have now chosen the values of γ_i from a normal distribution such that the total trans-heritability explained by the SNPs for a gene is roughly maintained at 10-14%.

We have now included these details in our main manuscript (Section Methods: Simulation Model Lines 915-939 and Supplementary Section B1).

17. Any justification why some specific tissues are enriched in the differential expression analysis for a given phenotype, but not the other tissues? Is there any evidence that the tissues which are possibly more relevant for the phenotype are also being found to be enriched?

The differential expression analysis was used to investigate whether the selected target genes for a trait were enriched for genes that have substantially different expression levels in one tissue compared to the rest. Although the analysis is done using trans-eQTL mapping statistics in whole blood, there can be signatures of trans-regulation patterns of different tissues since blood is connective tissue. If a tissue is found to be enriched in the differential expression analysis, it means that the genes that are highly differentially expressed (upregulated or downregulated) in that tissue are also overrepresented in the selected target genes for a trait, indicating that there can be a possibility that the gene expression in the enriched tissue might have a potential role in the overall genetic architecture of the trait. **We have included this explanation in the Results (Lines 374-375).**

Since the traits we chose to show as examples are complex traits, we do expect numerous potentially distinct tissues to be enriched for the selected target genes. However, our results show that several relevant tissues are enriched for the example traits. We found colon tissues to be enriched for differential expression of the selected target genes for ulcerative colitis (**Supplementary Figure 8**) and similarly several brain tissues are enriched for schizophrenia (**Supplementary Figure 6**).

18. Why the associations for genes 5 and 6 arise due to pleiotropy in the figure presenting gene network in the simulation study? More explanation is needed.

The association of genes 5 and 6 were due to the effects mediated through Gene 1 and 4 respectively. In comparison to Gene 5-6, Genes 7-9 have multiple independent associations with the SNPs. ARCHIE aims to identify such downstream genes that have evidence of multiple independent associations. Thus, to differentiate Genes 7-9 from Genes 5-6, we used the word pleiotropy, since the effects of SNPs 1-5 are cascaded to both Gene 7 and Gene 5 via Gene 1 (similarly for Gene 6 as well).

We have extensively modified and enhanced the section on simulations. Currently, in the simulation scenarios, we do not have such genes. Hence, we have omitted this explanation in the revised version of the manuscript.

19. In the main methods section of the paper, motivation of the method starting with individual-level data need to be included.

We have now amended the methods section of the paper to motivate the formulation of the sCCA framework from individual-level data.

Section Methods. Lines 811-831.

20. There are many typos in the paper.

We have checked the revised manuscript thoroughly to eliminate typos and grammatical errors.

Reviewer #2 (Remarks to the Author):

Review of Dutta et al. - Aggregative trans-eQTL analysis detects trait-specific target gene sets in whole blood

In this study, Dutta and colleagues develop a novel method to identify trait-specific trans-eQTLs and, as a result, prioritize trait-relevant genes. The authors use several publicly available datasets to show that it is possible to identify genes that were not found before using regular trans-eQTL mapping and they illustrate the relevance of the novel prioritized genes with several enrichment and follow-up analyses. The methodology of this paper is clever and it is a useful contribution on top of existing methods. However, the manuscript and in particular the figures could use editing for clarity and to emphasize the main message of the article. I also have some questions regarding the novelty of the results and some methodological details.

We thank the reviewer for the positive comments.

Major comments

1. Fig 1A could use more information and description within the figure panels, e.g., it is not immediately clear what is contained in the heatmaps (trans-eQTL summary Z-scores? P-values?). The flow of the diagram would be better if it went from left to right and/or top to bottom with a panel in between the 2 heatmaps that outlines the things ARCHIE does ('selects a subset of variants and genes') and data it uses (the LD matrix and co-expression from external datasets). You could even use Supp Fig 1 for this.

We have now modified Figure 1 according to the reviewer's suggestion. Current Figure 1 includes (A) properly labeled conceptual diagram for ARCHIE (B) an overall description of the data analysis. We have now included Figure 2 describing and reporting the simulations separately.

2. In the section 'Identification of downstream genes' please start by explaining that you are simulating a network with 9 genes and which of those you considered causal (all 9? Only 5-9?). Are genes 1-4 supposed to be local genes? Please also describe the difference of genes 5 and 6 as opposed to genes 7-9; I suppose the point is that they are all downstream effects but only 7-9 are affected by convergence of multiple significant genes, but that is not clear from the text. Also, 9 genes are a pretty low number for a set of causal genes, it would be good to simulate with more genes.

Currently, we have reformulated our simulation studies and have included a more diverse spectrum of alternate causal regulatory models for them (see our responses to comment 1 of reviewer 1). The current simulation studies have been conducted with 17-19 causal genes (8 cis/local genes and 9-11 distal/downstream genes). Although the number of genes is still low, we feel that this gives us explicit control over the causal flow of effects and allows us to better understand the properties of the method.

3. Were the novel genes identified by ARCHIE, e.g., the 59 for schizophrenia, tested in eQTLGen? And if so, do the ARCHIE genes and the non-significant eQTLGen trans-eQTLs have the same direction of effect?

Yes. By design, we include the genes that were tested in eQTLGen. Hence, the novel genes reported for each of the diseases were tested and reported in eQTLGen, although they had trans-eQTL mapping p -value $> 1 \times 10^{-06}$ across all the variants associated with the trait. We have now made that clear in the text:

Section Results. Line 283-288.

“The remaining 50.7% genes (termed “novel genes”) harbors only weaker ($0.05 > p$ -value $> 1 \times 10^{-06}$) associations and hence cannot be detected by standard trans-eQTL mapping alone; these genes display a similar pattern of trans-association with corresponding selected trait-related variants and are detectable only via the significant ARCHIE components. In fact, a large majority of the novel genes (89%) harbor multiple weaker trans-associations with the variants selected in the variant component.”

ARCHIE components estimate the regularized coefficient for an SNP or gene to be selected. So, the directions of the non-zero ARCHIE coefficients for the selected SNPs or genes might not be directly comparable with the direction of the standard trans-mapping effect size. Despite that, we compared the direction of the non-zero coefficients of SNP and gene components, with that of the standard trans-eQTL mapping coefficient. Across the three example traits, we found approximately 79% concordance of effect size direction, which indicates that the novel genes identified by ARCHIE capture correct regulatory effects.

4. How many of the novel ARCHIE genes are identified as being correlated to the polygenic risk score for the same trait? The eQTLGen consortium provides such correlation estimates (‘eQTS’) which sound like they should give similar results to your ARCHIE genes.

The reviewer points out an interesting aspect that requires further explanation. According to the suggestion of the reviewer, we performed eQTS analysis for the 59 novel genes identified for Schizophrenia (SCZ). Out of the 59 genes, 38 (64.4%) had nominal association evidence (p -value < 0.05) with at least one of the reported PRS for SCZ.

Section Results. Lines 334-343.

“Further, eQTLGen consortium reports the association of gene expressions with several publicly available polygenic risk scores (PRS) of SCZ at different p -value thresholds, curated from external databases. We found that out of the 59 novel genes identified, 38 (64.4%) had a nominal association with at least one reported PRS for SCZ.”

However, we would like to point out a key difference in the overall objective of ARCHIE with the eQTS analysis. In general, the association test of a polygenic risk score (PRS) for a trait with a gene expression (eQTS) assumes that a large majority of the SNPs included in the PRS have an effect on the gene expression proportional to their GWAS effect size on the

trait and in the same direction. This can be restrictive in terms of identifying subsets of SNPs that may have distinct regulatory effects through different pathways and gene networks. Through ARCHIE, we precisely aim to identify such patterns and such an approach can help to partition the overall PRS for a disease into distinct components according to underlying common pathways.

5. Were the genes that you specifically highlight in the text found in eQTLGen already or not? (CXCR4, DISC1, GPA33, NFKB1, etc.)

All of the genes we study in the paper are included in the eQTLGen database as they report summary statistics for trans-association statistics for ~10K trait-associated SNPs across all genes. However, the p-value for the genes we declared as “novel” detections via ARCHIE, had a p-value $> 1 \times 10^{-06}$ for all the SNPs associated with a given trait. These genes were not highlighted as significant findings in the eQTLGen manuscript. Our type-I error simulation studies show that a p-value threshold of 1×10^{-06} is needed for standard trans association analysis to maintain the type-I error rate at the desired nominal level of 0.05, although under the global null hypothesis. Thus, we believe our criterion for detection of “novel” gene in ARCHIE compared to standard trans analysis is reasonable.

As an example, the genes CXCR4 and CAV1 were novel identifications of ARCHIE for Schizophrenia (SCZ), which means that none of the 218 SNPs associated with SCZ tested in eQTLGen and included in the current analysis, had a strong trans-association with these genes. We have added the following text to clarify.

Section Results. Lines 302-303.

“These novel genes were not identified using traditional trans-eQTL mapping and were not reported as significant findings by the eQTLGen consortium.”

6. Where were the SNPs located on the genome? E.g., were the 9 variants for prostate cancer spread across multiple chromosomes or are they located in one or two loci?

The SNPs selected via the SNP component of ARCHIE were spread across multiple chromosomes and loci. For example, the SNPs that were selected by the two SNP components for prostate cancer were spread across 14 different loci. We have now added the chromosome and positions (GRCh 38) of the selected SNPs in Supplementary Table 2 and pointed this out in the manuscript as well.

Section Results: Prostate Cancer. Line 451.

7. You mention in the discussion that the direction of the effect of the SNPs on the trait is not incorporated into ARCHIE. Why not? It seems that the results would become much more interpretable if you use this information, because right now you aggregate trans-eQTLs from both risk and protected variants into your model. For simplicity, could you just subset the risk variants and run ARCHIE on those SNPs?

We thank the reviewer for this important suggestion. We agree that incorporating the relation between the variant and the trait being analyzed can potentially make the results more powerful and interpretable. However, as mentioned in response to Reviewer 1 Point 5, incorporating GWAS effects of the SNPs, especially the directions, within the ARCHIE analysis framework warrants further research.

eQTLGen provides the GWAS effect size and direction of the disease-related variants. According to the suggestion of the reviewer, we redid the ARCHIE analysis for 218 SNPs associated with Schizophrenia (SCZ) separately for the risk (112 SNPs) and protective (106 SNPs) variants respectively. For the 112 risk variants we selected 84 genes being trans-regulated by a subset of 23 variants. For the 108 protective variants, we selected 72 genes being trans-regulated by a subset of 29 variants. Overall, across the two categories, we found 61 genes to be common with the selected target genes in the full analysis with 218 variants.

However, we want to note that, reducing the number of variants would reduce the trait-specificity of the genes selected by ARCHIE, as shown in the simulation and resampling analysis (See Results section: Assessing trait specificity). Furthermore, the direction of association of a variant with the trait is an artefact of the allele being tested. By flipping the tested alleles of the protective variants and adjusting the sign (direction) of the trans-eQTL Z-values accordingly, we can essentially convert each variant into a risk-inducing variant. This, hence, would need further careful study, especially for interpretability.

Section Discussion. Lines 527-533.

“However, it is likely that incorporation of the GWAS effect sizes (value and direction) of trait association for the SNPs in the sCCA itself will lead to improved power for detection of the trait-specific target genes. One of the immediate ways to incorporate that might be to weight the variants with weights proportional to squared (or absolute value) of the GWAS effect size. However, incorporating the direction of the GWAS effect with the sCCA framework remains an interesting problem and merits further research. Additionally, incorporation of information on cis-genes and known functional annotation of genetic variants can improve the power of the analysis as well.”

8. Is there a way to include cis-eQTLs in ARCHIE? In figure 1B, the assumption is that gene expression levels of genes 5-9 are (partially) mediated by the gene expression levels of cis-genes 1-4. Could that increase the power to detect causal genes? Especially since some of the genes in the examples are located within GWAS loci.

We thank the reviewer for directing us to this important and interesting question. The reviewer is referring to the causal regulatory network used for simulation but is now no longer included in the manuscript. However, the general question still applies to the current simulation models that we considered. It is indeed true that the simulations are performed under the assumption that the trans-associations arise due to the partial mediating effect of the cis (local) genes on the distal genes.

One of the immediate ways to include cis-eQTLs in the method would be the following stepwise approach:

(1) Using the effect of the variant on cis-Gene (from existing transcriptomics studies) and the effect of the variant on the distal gene (from eQTLGen), estimate the effect of the local gene on the distal gene.

(2) Now use the estimated mediation effect matrix across all local (cis) genes and distal genes perform sCCA or other biclustering methods to identify a subset of local genes having effects on a subset of distal genes. If a trans-association is not substantially mediated by cis-genes, this analysis will down-weight those genes from the subsequent clustering analysis.

Thus, we believe our approach can be easily adapted to incorporate information on cis-mediation, but a detailed analysis of the power of this approach is beyond the scope of the current manuscript. We would, however, like to point out that such as approach addresses the cis-mediation hypothesis and identifies gene networks where effects of GWAS variants are being cis-mediated significantly to a network of distal genes. The current version of ARCHIE, on the other hand, can capture associations arising due to any mechanism which includes cis mediation and other mechanisms like chromosomal topology.

We have now noted in Section Discussion. Lines 527-533 that the power of ARCHIE can be potentially improved by incorporating additional information, including GWAS effect-sizes and information cis-genetic regulations, and future research is merited to further explore these areas.

9. What is the effect of using a co-expression matrix from a reference panel (GTEx)? Why did you not use the co-expression from the eQTLGen consortium directly?

We agree with the reviewer that a higher accuracy could have been achieved with the co-expression matrix estimated within eQTLGen. Unfortunately, eQTLGen does not provide individual expression level data to compute the co-expression matrix. Further, since we claim that our method can work on summary data, using a coexpression matrix from GTEx reinforces that claim. As a response to Reviewer 3 Point 2, we have now shown that the results are robust to the choice of studies for coexpression matrix, by constructing the coexpression matrix from gene expressions in whole blood in the DGN study and comparing the results.

10. “we found the selected trans-associations, in particular, the target genes mediate significant trait heritability (p -value < 0.001) than expected by chance” this sentence needs a re-write to become understandable. The enrichment of trait heritability is a nice method that, if I am not mistaken, is novel. I think you can emphasize this method and its results more in the main text. Also, figures 2-4D would be stronger if you add the estimated trait heritability using significant trans-eQTL SNPs for both the trait under investigation (e.g., SCZ, UC, prostate cancer) as well as another trait (or the results of the resampling method), which would be lower than the ARCHIE selected SNPs. Right now, the message of this picture seems to be: ‘ARCHIE selected SNPs to contribute less to heritability than all SNPs associated to this trait’, while I suppose that you want it to say: ‘ARCHIE selected SNPs are specifically enriched for heritability of the trait we are

investigating’.

Thank you for pointing this out. Indeed, the estimation of trans-heritability is a novel component in our follow-up analysis. We have now tried to emphasize and better explain the intuition and results from this analysis within the main text.

Section Results: Schizophrenia. Lines 315-321.

“Next, we tested whether the selected variants and genes were enriched in trans-heritability for SCZ. Using an expression imputation approach (See Methods for details), we estimated the approximate heritability of SCZ explained by the trans-associations between the selected genes and variants, using individual level data from UK Biobank. We compared the estimate to (1) the expected distribution of heritability explained by the selected SNPs and randomly chosen genes (excluding the selected genes) for SCZ and (2) the expected distribution of heritability explained by the selected SNPs and genes for a randomly chosen trait. The results showed that the selected SNPs and genes are significantly enriched in trait heritability (p -value < 0.001) than expected by chance (Figure 3D).”

Section Results: Ulcerative Colitis. Lines 391-394.

“Further, similar to SCZ, we found the associations of the SNPs with target genes was strongly enriched (p -value < 0.001) for heritability of UC than expected by chance alone and also the selected trans-associations explained heritability of UC more than expected for a random trait or disease (Figure 4D).”

Section Results: Prostate Cancer. Lines 453-455.

“Additionally, similar to SCZ and UC, we found evidence of enrichment of trans-heritability of PC that can be mediated by the target genes and also that the trans-heritability mediated by the selected SNPs and genes was significantly more than that for a randomly chosen trait (Figure 4D), but the level of significance achieved was relatively weaker (p -value = 0.001 and 0.007; See Methods for details).”

We also appreciate the reviewer’s suggestion on the figures. We have now altered the numerical experiments and figures according to the suggestion. In Figures 3-5D (previously Figures 2-4D), we now show two violin plots each (See Figure below). The left violin plot corresponds to the pseudo r^2 of TIES (estimated trans-heritability) for a random set of genes of the same number as selected in the gene component with the same trait. Thus, it reflects the distribution of trait-heritability of the SNPs mediated via random genes. The right violin plot corresponds to the pseudo r^2 of TIES (estimated trans-heritability) for the selected SNPs and genes with a randomly chosen trait in UK Biobank, such that the original trait in analysis had low coheritability. Thus, it reflects the amount of trait heritability of selected SNPs mediated via the selected genes. The estimated trans-heritability for the selected genes and SNPs for a given trait is given by the horizontal red dashed line.

From the figures we see that, the trans-heritability mediated by the genes selected by ARCHIE is substantially higher than any random set of genes and further the trans-heritability of the selected SNPs and genes is also higher than that of any random trait, indicating that the selected SNPs and genes jointly might potentially be trait specific.

Minor comments

1. Although I understand the desire for a catchy name, ‘ARCHIE’ is far from an acronym. At this point, I suggest to either just call it ‘ARCHIE’ without highlighting random letters in the words or to come up with an actual acronym.

We appreciate the comment. We have referred to the method as “ARCHIE” throughout, without highlighting the letters in the constituent words.

2. There are quite a number of mistakes in verb conjugation, e.g. “ARCHIE selects the downstream genes (Genes 7-9) that has trans-associations” should be “ARCHIE selects the downstream genes (Genes 7-9) that have trans-associations”. Please have a careful look at the grammar throughout.

We have now thoroughly checked the sentences for grammatical errors.

3. “The selected genes are broadly trans-regulated by the selected SNPs and mediate their effect to the trait.” This statement is unsubstantiated. You cannot be sure that the genes mediate the trait without testing for causality or mediation analyses.

We agree with the comment. We have rephrased this sentence as:

“The selected genes reflect gene sets that are broadly trans-regulated by the selected SNPs and via which the effects of selected SNPs on the trait are appear to be mediated”

Section Results. Line 109-110.

4. “we found that ARCHIE selects the downstream genes (Genes 7-9) that has trans-associations to multiple upstream genetic variants, with high probability” ◊ please add the actual probability here.

We have now modified the simulation setup and currently consider several causal regulatory models. Although the actual power, specificity and sensitivity estimates might vary depending on the choices of the simulation parameters, we believe the relative positioning of estimates for ARCHIE as compared to that of standard trans-eQTL mapping is visually more instructive and provides the main message of the simulation studies. However, we have documented the estimated power for sparse and dense causal network models as shown in Figure 2, in Supplementary Table 1.

5. What was the rationale for using sample sizes of 1,000 and 30,000 in the simulation studies?

The choice of the sample sizes for simulation was to roughly reflect the sample sizes for GTEx (v8) and eQTLGen studies, which we are widely used transcriptomic studies and has been referred to several times in our manuscript as well. We have updated the text to reflect this.

Section Results. Lines 145-147.

“Each causal scenario was simulated at two sample sizes ($N = 1,000$ and $30,000$) reflecting the approximate sample sizes of GTEx v8 and eQTLGen studies respectively.”

6. How did you select the 3 traits with more in-depth results?

The example traits were selected such that they represent potentially different categories: Neuropsychiatric, autoimmune/gastrointestinal and cancer. We have updated the text to reflect this.

Section Results. Line 289-291.

“Here, we focus on results for three different phenotypes, reflecting three different classes of diseases, their corresponding trans-association patterns, the selected target gene-sets and the novel genes detected by ARCHIE.”

7. Please briefly describe the ‘competitive null distributions’ in the legends for figures 2, 3 and 4.

We have now included this in the figure legends (now Figures 3, 4 and 5).

8. Please indicate in figures 2-4A of the ARCHIE components which ones are significant, and which are not.

Currently, the Figures being referenced here are Figure 3-5A. We have added an asterisk (*) sign in the Figures next to the components which are selected for the traits to indicate significance/selection.

9. Please add a legend for the colors in Fig. 2C and 3B. It is also a bit confusing to plot 2 variables on the x-axis, it would be clearer if you plot only the p-value and color the bars by % overlap or vice versa.

We have added a legend for the bar plots (for p-value and proportion overlap) corresponding to the pathways to make the reading clearer.

10. Please label the y-axis of Fig. 3C.

We have added this.

11. Please label the axes of Supp Fig 2

We have added this.

Reviewer #3 (Remarks to the Author):

Review:

Aggregative trans-eQTL analysis detects trait-specific target gene sets in whole blood

Overview:

Diptavo et al. introduce a new method called Aggregative tRans assoCiation to detect pHenotype specific gEne-sets (ARCHIE) to detect the trans associations between a set of known trait-related genetic variants and a set of co-regulated genes. This method is motivated by the omnigenic model which predicts that the majority of SNP-heritability is mediated by pervasive trans-regulatory effects, which are historically challenging to identify due to limited statistical power. The core inferential framework of ARCHIE is sparse Canonical Correlation Analysis coupled with a permutation test to identify trans-regulatory networks limited to a set of known trait-associated genetic variants. A key novelty is the ability of ARCHIE to run directly on GWAS and eQTL summary statistics, provided estimates of population LD and gene co-expression from publicly available datasets (1000G, GTEx, etc). Applying the ARCHIE model GWAS summary statistics from 29 traits together with eQTL summary data from eQTLGen, they identify many novel genes which are supported with gene set enrichment/ontology analyses.

Overall, I found the paper to be well written, highly relevant, and the statistical methodology sound. With that said, I have a few comments related to simulations and real-data replication.

We would like to thank the reviewer for their positive comments.

Major Comments:

1. I appreciate the use of simulations to demonstrate the feasibility of ARCHIE under ideal circumstances and thank the authors for their inclusion, however, I have three primary concerns that could be addressed with expanded or altered simulations.

a. First, I find the parameters used to be a bit arbitrary and likely unreflective of actual biology. Can the authors re-parameterize cis- and trans-effect sizes to be proportional to cis-h²_g or trans-h²_g estimates? For example, if the cis-h²_g of a given upstream/core-gene is 0.1 and there are 5 SNPs with direct effects, then (assuming no LD) $\beta \sim N(0, 0.1 / 5)$. If trans-regulatory effects are due to distal cis-effects, then the mediating effect can be 'reverse-engineered' from the cis-h²_g at the cis-gene(s). This would help place findings and statistical power into a realistic context.

We agree that the choice of simulation parameters was relatively arbitrary. We thank the reviewer for pointing that out. We have substantially amended the simulation section and constructed newer simulation scenarios to better and accurately characterize the properties of our method ARCHIE (See response to Reviewer 1 and 2 for further details).

According to the suggestion of the reviewer, we have now made the choices of parameters such that it reflects the results from current studies. The cis-regulatory effects of SNPs on cis

gene expressions were chosen such that the average cis heritability explained by the SNPs were maintained at 20-22%. The regulatory effects between genes in causal networks (Figure 2B, C and Supplementary Figure 2) were chosen such that the total trans-heritability of the expressions of Gene 1-9 explained by the SNPs were maintained at 10-14%.

We have now made this clearer in the simulations model (Lines 123-185 and Lines 885-907) and Supplementary Methods Section B1.

b. Second, the only simulation performed was under the alternative where trait-specific trans-regulatory action is occurring on genes measured in the relevant tissue. Adding simulations under the null demonstrating that p-values are well-calibrated would greatly help interpretation. There is some numerical evidence presented to this using real-data results, but a clearer link to calibration and QQ plots would be helpful.

We have now added a detailed simulation under the global null scenario and estimated the empirical type-I error of ARCHIE and compared that to standard trans-association analysis. See Response to Reviewer 1 for details. Further, we have simulated to evaluate type-I error and power under a competitive null hypothesis where the goal is to detect trait-specific patterns. See Main Text line 187-242 for details regarding these simulations and Supplementary Methods Section B.

c. Last, it would be helpful to see how ARCHIE performs in simulations with model misspecification regarding the use of external Sigma_{EE} matrices. For example, the regulatory/generative model could be fixed to generate Sigma_{GG}, Sigma_{GE} matrices, but a noisy (sampling around some true Sigma_{EE} using Wishart) or incorrect Sigma_{EE} (relevant regulatory networks are only partially captured/correlated due to “non-relevant” tissue) is used in the inference procedure.

Since for most of the applications, the coexpression matrix (Σ_{EE}) is not positive definite matrix (sample size < number of genes), it would be challenging to simulate such matrices from the Wishart distribution. However, in the spirit of the reviewer’s comment, we investigated the robustness of the ARCHIE results by varying the sample size of the reference transcriptomic study and studying the power of ARCHIE. The sample size for trans-eQTL summary statistics was set to 10,000. In the attached Figure (also included as Supplementary Figure 9), we varied the size of the reference data from 100 to 700. Our results show that beyond an approximate sample size of 450, the power of ARCHIE remains constant with the increasing sample size of the reference data. However, the optimal sample size depends on the sample size of the standard trans-eQTL study as well, which in this case was set to 10,000. In general, it has been shown that a plateau is reached in terms of power for an optimal ratio of the sample sizes of trans-eQTL study and reference transcriptomic study⁵.

We have included a detailed description of this experiment in Supplementary Section B5 and the Discussions (Line 488-490).

“ARCHIE involves estimating LD and coexpression from publicly available reference datasets. However, we found that the results can be robust to the choice of reference data and estimation errors arising due to sampling variations (Supplementary Figure 9). In general, robust performance can be achieved under an optimal ratio of the sample sizes of the reference transcriptomic study and the study from which summary statistics of standard trans-eQTL associations are being analyzed.”

2. I'm glad the authors performed several enrichment analyses to provide support for their gene lists, however, no thorough perturbation check was performed on using publicly available data as a means to estimate the correlation matrices. Are the real-data results robust to replacing Sigma_EE estimated in GTEx whole blood expression with say, whole blood expression from DGN (at least within genes overlapping both studies) or some reasonable alternative?

We thank the reviewer for this important point. We have now included an example analysis for Schizophrenia (SCZ) using a coexpression matrix constructed from the genes in Depression Genes and Network (DGN) study.

Overall, the results show a high degree of overlap for the selected target genes in our original analysis using coexpression constructed from GTEx v8 and that from DGN. Approximately 83% of the selected genes in GTEx v8 were also selected in the analysis using coexpression from DGN. In particular, 76% of the novel genes were also captured. This indicates that the selection results from ARCHIE can be potentially robust to the choice of the reference transcriptomic dataset. We have pointed this out in the results (Line: 303-313) and in Supplementary Methods Section C3.

“We further investigated the robustness of the results using coexpression estimates from gene expressions levels reported in Whole Blood by the Depression Genes and network (DGN) study.

The results show that out of 75 genes and 59 novel genes identified by ARCHIE, 62 and 45 were also selected in the replication analysis using data from DGN, demonstrating the potential robustness of the results by ARCHIE.”

Minor Comments:

1. What is the alpha threshold used to determine power for simulations/Figure 1C?

We have now calibrated the type-I error rates for standard trans-eQTL mapping and ARCHIE and reported the corresponding simulation results. In general, the level used for standard trans-eQTL is substantially lower than ARCHIE to maintain a similar type-I error rate since ARCHIE is highly conservative at a given level (See response to Reviewer 1 Point 1 for further details).

2. Why was a significance threshold of 0.001 used in Supplementary Figure 2?

Since we are using a resampling-based method, we can only detect p-values up to a minimal threshold depending on the number of resampling iteration. We have now increased the iterations to 100,000 for results presented in Supplementary Figure 4 (previously Supplementary Figure 2) that guarantees that we can reliably estimate the p-value of the order of 0.0001 or higher.

3. The regression formula at the bottom of page 27 seems to have a typo. Should there be ‘beta_j’ for each of the G TIES(p) values?

Thanks for catching this typo. We have corrected this.

1. Feder, A. F., Petrov, D. A. & Bergland, A. O. LDx: Estimation of Linkage Disequilibrium from High-Throughput Pooled Resequencing Data. *PLoS One* **7**, e48588 (2012).
2. Schäfer, J. & Strimmer, K. A shrinkage approach to large-scale covariance matrix estimation and implications for functional genomics. *Stat. Appl. Genet. Mol. Biol.* (2005). doi:10.2202/1544-6115.1175
3. Khandaker, G. M. *et al.* Inflammation and immunity in schizophrenia: implications for pathophysiology and treatment. *The Lancet Psychiatry* **2**, 258–270 (2015).
4. Zhao, P. & Yu, B. On model selection consistency of Lasso. *J. Mach. Learn. Res.* (2006).
5. Kundu, P., Tang, R. & Chatterjee, N. Generalized meta-analysis for multiple regression models across studies with disparate covariate information. *Biometrika* (2019). doi:10.1093/biomet/asz030

REVIEWERS' COMMENTS

Reviewer #1 (Remarks to the Author):

The authors have addressed my comments adequately.

Reviewer #2 (Remarks to the Author):

I would like to thank the authors for addressing my comments thoroughly and thoughtfully. The figures and manuscript have definitely improved and I am happy to accept the work. I just have one minor comment: Figure 2B and 2C are exactly the same.

Reviewer #3 (Remarks to the Author):

The authors have performed a considerable amount of work addressing my comments. I have no further points at this time.